# A Fair Bayesian Inference through Matched Gibbs Posterior

**Jihu Lee[1], Kunwoong Kim[2][†], Sehyun Park[1], Insung Kong[3], Dongyoon Yang[4], Yongdai Kim[1][◇]**

[1]Department of Statistics, Seoul National University
[2]KAIST
[3]Department of Applied Mathematics, University of Twente
[4]AI Advanced Technology, SK Hynix Inc.
`{superstring1153, kwkim.online}@gmail.com, ps_hyen@snu.ac.kr,`
`insung.kong@utwente.nl, dongyoon.yang@sk.com, ydkim0903@gmail.com`

## Abstract

With the growing importance of trustworthy AI, algorithmic fairness has emerged as a critical concern. Among various fairness notions, group fairness - which measures the model bias between sensitive groups - has received significant attention. While many group-fair models have focused on satisfying group fairness constraints, model uncertainty has received relatively little attention, despite its importance for robust and trustworthy decision-making. To address this, we adopt a Bayesian framework to capture model uncertainty in fair model training. We first define group-fair posterior distributions and then introduce a fair variational Bayesian inference. Then we propose a novel distribution termed matched Gibbs posterior, as a proxy distribution for the fair variational Bayesian inference by employing a new group fairness measure, the matched deviation. A notable feature of matched Gibbs posterior is that it approximates the posterior distribution well under the fairness constraint without requiring heavy computation. Theoretically, we show that the matched deviation has a strong relation to existing group fairness measures, highlighting desirable fairness guarantees. Computationally, by treating the matching function in the matched deviation as a learnable parameter, we develop an efficient MCMC algorithm. Experiments on real-world datasets demonstrates that matched Gibbs posterior outperforms other methods in balancing uncertainty–fairness and utility–fairness trade-offs, while also offering additional desirable properties.

## 1 Introduction

Artificial intelligence (AI) technologies have been incredibly successful and are now widely used as essential decision-making tools in a variety of fields, such as college admissions, criminal risk assessment, and credit scoring. However, when observed data contain unfair biases, the resulting trained models may have biases favoring specific groups, such as white individuals or males (Angwin et al., 2016; Ingold & Soper, 2016; Dua & Graff, 2017), which leads us to consider algorithmic fairness in AI-based decisions as a crucial mission (Corbett-Davies et al., 2017; Starke et al., 2022).

Among various notions of algorithmic fairness, *group fairness* is the most widely studied, which requires that certain statistics across protected groups remain similar. For example, the ratio of positive predictions should be similar across each protected group (Calders et al., 2009; Barocas & Selbst, 2016; Zafar et al., 2017; Donini et al., 2018; Agarwal et al., 2018b). In turn, a large amount of research have been conducted to develop algorithms for ensuring group fairness in various supervised learning tasks (Zafar et al., 2017; Donini et al., 2018; Agarwal et al., 2018b; Quadrianto et al., 2019; Jiang et al., 2020b).

---

[†]Work done while at Seoul National University.
[◇]Corresponding author.

Existing algorithms for group fairness focus on finding an accurate prediction model under fairness constraints. However, learning deep neural networks (DNNs) which are highly over-parameterized and so susceptible to overfitting to lead over-confident predictions. Proper quantification of uncertainties in prediction of DNNs becomes a **crucial mission** to make AI algorithms more reliable and trustworthy (Gal et al., 2016; Malinin & Gales, 2018; Mariet et al., 2021). For example, performing medical decision making such as disease diagnosis, uncertainty quantification is essential for supporting a physician's decision making while the datasets are biased with respect sensitive attributes; e.g. skin cancer diagnosis with biased dataset (Wei et al., 2024), Alzheimer's disease diagnosis with biased dataset (Bercea et al., 2023).

Bayesian inference turns out to be a useful approach for this purpose due to its ability to quantify uncertainties in prediction models (Neal, 2012; Gal & Ghahramani, 2016; Kendall & Gal, 2017; Abdar et al., 2021; Gawlikowski et al., 2023). For example, McAllister et al. (2017); Michelmore et al. (2020); Ding et al. (2021b) highlight the importance of uncertainty quantification (UQ) in autonomous vehicles, and suggest Bayesian deep learning to handle it. Bayesian inference is also well utilized in medical image segmentation, where UQ is considered important (Sedai et al., 2019; Yu et al., 2019; Wang et al., 2020; 2021; Shi et al., 2021; Wang & Lukasiewicz, 2022).

Nonetheless, Bayesian inference under group fairness has received little attention, which is the main theme of this paper. Modifying the single accurate fair prediction model for Bayesian inference would not be easy, since Bayesian inference needs a (posterior) distribution of prediction models. For fair Bayesian inference, we first propose the definition of group fairness of a posterior distribution[1]. Then, we develop a variational inference (VI) method for searching a group-fair posterior distribution approximating the full Bayesian posterior distribution well without heavy computation. Note that a simple modification of the standard VI, such as *mean-field Gaussian model* (Graves, 2011; Blundell et al., 2015) for group fairness, would not be practically feasible due to the computational burden of calculating the group-fair constraint at each gradient update.

We resolve this problem by employing the idea of Gibbs posterior, which is a useful tool to quantify uncertainties in a prediction model learned by minimizing an objective function (e.g., the empirical risk) (Zhang, 2006; Jiang & Tanner, 2008; Bissiri et al., 2016; Grünwald & Mehta, 2020; Miller, 2021; Martin & Syring, 2022; Syring & Martin, 2023). It provides a posterior distribution by treating the objective function as the log-likelihood. Gibbs posterior is popularly used when specifying the likelihood is difficult, with applications of clustering (Rigon et al., 2023), PCA (Winter et al., 2023), and medical image registration (Wang et al., 2022). The main contribution of this paper is to propose a specially designed penalized log-likelihood for group fairness and develop an efficient MCMC algorithm for the corresponding Gibbs posterior. By the proposed Gibbs posterior, we can explore the posterior distribution without any constraint on the parameter space and thus standard MCMC algorithm can be directly used without much hamper. By numerical experiment, we illustrate that the proposed Gibbs posterior outperforms the mean-field Gaussian VI and other baseline methods, in terms of prediction accuracy and uncertainty quantification.

We can summarize our contributions as follows:

- To enable uncertainty quantification of group-fair models, we consider variational Bayesian approaches. We define the level of group fairness of a given posterior, and propose variational Bayesian inference methods under the group fairness.
- We propose matched Gibbs posterior, a novel distribution based on the penalized log-likelihood with the matched deviation as a penalty for group fairness, to successfully perform the fair variational inference. In addition, we provide a novel theoretical relationship between the matched deviation and other group fairness measures, which explains the benefit of employing the matched deviation in controlling group fairness.
- We empirically demonstrate that the proposed matched Gibbs posterior outperforms baseline methods by analyzing multiple real-world datasets, in terms of trade-off between utility and group fairness as well as trade-off between uncertainty and group fairness. We show that matched Gibbs posterior also has a desirable property of improved individual fairness as a by-product.

---

[1]We mean by 'a posterior distribution' any distribution on the parameter space. For the posterior distribution proportional to the product of the likelihood and the prior, we call it 'the full Bayesian posterior'.

## 2 Preliminaries

Let $Z = (X, Y, S)$ be a random vector, where $X \in \mathcal{X}$ is the input, $Y \in \mathcal{Y}$ is the output and $S \in \{0, 1\}$ is the sensitive attribute. Let $z_i = (x_i, y_i, s_i), i = 1, \ldots, n$ be given data, which are independent copies of $Z$, and we write $\mathcal{D}_n = \{z_i\}_{i=1}^n$. Let $\mathcal{F}$ be the set of prediction models $f(x, s)$ where the domain is $\mathcal{X} \times \{0, 1\}$ with the codomain being $\mathbb{R}$ for regression problems and $\{p \in [0, 1]^c : \|p\|_1 = 1\}$ for $c$-class classification problems (i.e., $\mathcal{Y}$ are $\mathbb{R}$ and $\{1, \ldots, c\}$, respectively). For given $f$, let $\ell(f)$ be the log-likelihood of $f$ given $\mathcal{D}_n$.

### 2.1 Notion of group fairness

Three representative criteria for group fairness are independence, separation and sufficiency (Castelnovo et al., 2022; Barocas et al., 2023). In this paper, we focus on independence, so-called Demographic Parity[2]. The formal definition of demographic parity, often called as a statistical parity, is as follows (Agarwal et al., 2018a).

**Definition 2.1** (Demographic Parity, DP). A prediction model $f$ satisfies demographic parity under the distribution of $Z = (X, Y, S)$, if $f(X, S) \perp S$ holds. In other words, $\mathbb{P}_{f,0} \equiv \mathbb{P}_{f,1}$, where $\mathbb{P}_{f,s}$ is the conditional distribution of $f(X, s)$ given $S = s$.

From the above definition, we can naturally define the demographic parity gap as follows. Let $\psi(\cdot, \cdot)$ be a given deviation measure of two probability measures.

**Definition 2.2** (DP gap). The ($\psi$-)demographic parity (DP) gap of a prediction model $f$ is defined as

$$\Delta_\psi(f) := \psi(\mathbb{P}_{f,0}, \mathbb{P}_{f,1}). \tag{1}$$

The empirical $\psi$-DP gap of $f$ is defined as $\Delta_{n,\psi}(f) := \psi(\mathbb{P}_{f,0}^n, \mathbb{P}_{f,1}^n)$, where $\mathbb{P}_{f,s}^n$ is the empirical distribution of $\{f(x_i, s)\}_{i:s_i=s}$. We say that $f$ is $\psi$-group-fair with level $\delta$ if $\Delta_{n,\psi}(f) \leq \delta$, and we call $\Delta_{n,\psi}(f) \leq \delta$ the ($\psi$-)DP$_\delta$ constraint. For notational simplicity, in the followings we drop the subscript $n$ in $\Delta_{n,\psi}(f)$ and denote $\Delta_\psi(f)$ as the empirical $\psi$-DP gap unless there is any confusion.

Note that the original measure for the violation of DP (Agarwal et al., 2018a; 2019) is $\Delta_{\mathrm{DP}}(f) := |\Pr(f(X, 0) \geq \tau | S = 0) - \Pr(f(X, 1) \geq \tau | S = 1)|$ for a prespecified threshold $\tau$. However, dependency on the specific value of $\tau$ limits the reliability of this measure (Silvia et al., 2020). To resolve the limitation, the $\psi$-DP gap has been studied through line of research (Jiang et al., 2020a; Chzhen et al., 2020; Silvia et al., 2020; Barata et al., 2021). Corresponding examples of $\psi$-DP gaps are: (i) $\Delta_{\mathrm{W}}(f) := \mathcal{W}_2^2(\mathbb{P}_{f,0}, \mathbb{P}_{f,1})$, (ii) $\Delta_{\mathrm{TV}}(f) := \mathrm{TV}(\mathbb{P}_{f,0}, \mathbb{P}_{f,1})$, (iii) $\Delta_{\mathrm{KS}}(f) := \mathrm{KS}(\mathbb{P}_{f,0}, \mathbb{P}_{f,1})$. Here, $\mathcal{W}$, TV and KS is the Wasserstein distance, the total variation distance, and the Kolmogorov-Smirnov distance, respectively. See Appendix A.1 for the definitions of each. From now on, we slightly abuse notation and use $\Delta_\psi(f)$ and $\Delta_\psi(\theta)$ interchangeably, when $f$ is parameterized by $\theta$.

### 2.2 Variational Bayesian inference

For a given prior $\pi$ on $\mathcal{F}$, the full Bayesian posterior distribution is given as $\pi(f|\mathcal{D}_n) \propto \exp(\ell(f))\pi(f)$. When it is difficult to calculate (or generate samples from) $\pi(f|\mathcal{D}_n)$, a well known remedy is to search a distribution $\nu$ on $\mathcal{F}$ that approximates $\pi(f|\mathcal{D}_n)$ well and is easy to be handled (e.g., easy to generate samples), which is often called as variational inference (VI) (Graves, 2011; Blei et al., 2017). For VI, we first choose a variational class $\mathcal{V}$ of distributions, called variational distributions, which are easy to be handled. Then, we find the optimal variational distribution $\nu^*$ that minimizes the KL divergence between $\nu$ and $\pi(\cdot|\mathcal{D}_n)$ as stated formally:

$$\nu^*(f) := \arg\min_{\nu \in \mathcal{V}} D_{\mathrm{KL}}\big(\nu(\cdot)\|\pi(\cdot|\mathcal{D}_n)\big). \tag{2}$$

Finally, all of downstream Bayesian inference tasks such as uncertainty quantification and estimation of the predictive distribution are done with $\nu^*$ instead of $\pi(\cdot|\mathcal{D}_n)$. For example, the predictive distribution of $Y$ given $X = x$ is obtained by $p(Y = y|X = x) = \int p(Y = y|X = x, f)\nu^*(df)$.

---

[2] We extend our discussion and methodology to the Equalized Odds criteria. See Appendix E.1 for details.

# 3 VARIATIONAL BAYESIAN INFERENCE UNDER GROUP FAIRNESS

## 3.1 CHALLENGES IN THE FULL BAYESIAN INFERENCE

The posterior distribution under the $\mathrm{DP}_\delta$ constraint could be defined as the full Bayesian posterior $\pi(f|\mathcal{D}_n)$ truncated on the set of $\psi$ group-fair prediction models with level $\delta$, i.e., $\{f : \Delta_\psi(f) \leq \delta\}$. However, this constrained full Bayesian posterior is hard to compute, because we cannot formulate the constrained space analytically, especially when using complex prediction models such as deep neural networks. Also, since $\pi\{f : \Delta_\psi(f) \leq \delta|\mathcal{D}_n\}$ is usually very small, acceptance-rejection sampling is not practically feasible. See Appendix C.2 and Appendix C.3 for further explanations.

## 3.2 DEFINITION OF GROUP-FAIR POSTERIOR DISTRIBUTIONS

We say that a *given posterior $\nu$ is strongly $\psi$-fair with level $\delta$* if $\nu\{f : \Delta_\psi(f) \leq \delta\} = 1$. Our aim is to find a strongly $\psi$-fair posterior $\nu$ which approximates the full Bayesian posterior well. The definition of strongly $\psi$-fair posterior distributions, however, is very restrictive and hard to be satisfied. Even for linear models, variational inference with the mean-field Gaussian models could not be applied to find a strongly $\psi$-fair posterior since none of variational distributions are strongly $\psi$-fair. See Appendix C.5 for details.

A remedy is to relax the definition of fairness. We say that $\nu$ is $\psi$-fair with level $\delta$ when $\mathbb{E}_{f \sim \nu} \Delta_\psi(f) \leq \delta$. Here, we call $\mathbb{E}_{f \sim \nu} \Delta_\psi(f)$ the *average DP gap*. Our strategy is to find a good fair posterior $\nu^{(w)}$ whose average DP gap is upper bounded, and make it to strongly $\psi$-fair by the rejection sampling. That is, for a given fair posterior $\nu^{(w)}$ with level $\eta < \delta$, we obtain a strongly fair posterior $\nu^{(s)}$ by letting $\nu^{(s)}(\cdot) \propto \nu^{(w)}(\cdot)\mathbb{I}(\Delta_\psi(\cdot) \leq \delta)$. The Markov inequality implies that $\nu^{(w)}\{f : \Delta_\psi(f) \leq \delta\} > 1 - \eta/\delta$ and thus the rejection sampling is expected to work well. See Appendix C.5 for empirical evidences.

## 3.3 FAIR VARIATIONAL INFERENCE

We explain how to modify the standard VI for group-fairness. Suppose that $\mathcal{F}$ is parameterized by $\Theta \subset \mathbb{R}^d$. That is, $\mathcal{F} = \{f_\theta : \theta \in \Theta\}$ for $\theta := (\theta_1, \ldots, \theta_d)^\top$. For a proxy distribution on $\Theta$ in the variational inference, let $\{\nu(\cdot|\gamma), \gamma \in \Gamma\}$ be a given class of proxy distribution parameterized by $\gamma \in \Gamma$. Then, we search $\gamma$ which maximizes the ELBO under the constraint $\mathbb{E}_{\theta \sim \nu(\cdot|\gamma)} \Delta_\psi(f_\theta) \leq \delta$. As usual, we maximize the penalized ELBO formularized as: $\mathrm{ELBO}(\gamma) - \lambda n \mathbb{E}_{\theta \sim \nu(\cdot|\gamma)} \Delta_\psi(f_\theta)$, where $\lambda$ is the Lagrangian multiplier. Specifically, the ELBO term is defined as:

$$\mathrm{ELBO}(\gamma) := \mathbb{E}_{\theta \sim \nu(\cdot|\gamma)}[\ell(\theta)] - D_{\mathrm{KL}}\big(\nu(\cdot)||\pi(\cdot)\big). \tag{3}$$

## 3.4 CHALLENGES IN FAIR VARIATIONAL INFERENCE

When $\nu(\cdot|\gamma)$ is a tractable distribution such as a mean-field Gaussian model which assumes that $\nu(\cdot|\gamma)$ is a product of Gaussian distributions, the ELBO can be calculated easily: the first term can be calculated through samples generated from $\nu(\cdot|\gamma)$, while the second term is analytically tractable when the prior is also a product of Gaussian distribution.

A critical problem is to calculate the average DP since computation of $\Delta_\psi(f_\theta)$ is computationally demanding. In particular, when $\psi$ is the KL divergence or integral probability metrics (IPMs), computation of the average DP gap requires adversarial learning which is computationally demanding and numerically unstable. To be more specific, consider the IPM with a discriminator class $\mathcal{G}$ as $\psi$ defined as:

$$\Delta_{\mathrm{IPM}_\mathcal{G}}(f) := \sup_{g \in \mathcal{G}} \left| \int g(f(x, 0))\mathbb{P}_{n,0}(dx) - \int g(f(x, 1))\mathbb{P}_{n,1}(dx) \right|. \tag{4}$$

To calculate $\Delta_{\mathrm{IPM}_\mathcal{G}}(f)$ for each $f$, we have to find the discriminator $g_f$ that maximizes $\left| \int g(f(x))\mathbb{P}_{n,0}(dx) - \int g(f(x))\mathbb{P}_{n,1}(dx) \right|$, which is computationally demanding. In particular, when calculating the average DP gap, we have to find $g_{f_\theta}$ for all $\theta \sim \nu(\cdot|\gamma)$, which would be practically infeasible. One may argue that certain deviation measures such as MMD (Gretton et al.,

2012) do not require an adversarial learning, but the computation of MMD is still demanding when $n$ is large since the computational complexity is $O(n^2)$.

In the following section, we propose a specially designed Gibbs posterior approach to resolve the problem in fair VI. We first develop a novel deviation measure called matched deviation and then use the Gibbs posterior based on the log-likelihood penalized by the matched deviation as a proxy distributions of fair VI, which we call matched Gibbs posterior. Its computation is practically feasible since adversarial learning is not required and an efficient MCMC algorithm can be implemented whose computational complexity of each update is $O(n)$, instead of $O(n^2)$.

## 4 MATCHED GIBBS POSTERIOR FOR FAIR VARIATIONAL INFERENCE

A Gibbs posterior is a tool to obtain a posterior without specifying the full likelihood (Zhang, 2006; Jiang & Tanner, 2008; Bissiri et al., 2016). Suppose that we estimate $\theta$ by minimizing a certain objective function $R_n(\theta)$. Then, the Gibbs posterior with respect to $R_n$ and prior $\pi$ is defined as $\nu_n(\theta) \propto \exp(-R_n(\theta))\pi(\theta)$.

Our aim is to develop a class of objective functions $\{R_n(\theta; \lambda) : \lambda \in \Lambda\}$ to use the corresponding Gibbs posteriors $\nu_n(\cdot; \lambda)$ as the class of proxy distributions and estimate $\lambda$ by maximizing the ELBO under the fairness constraint.

In general, we estimate $\theta$ by minimizing the negative log-likelihood $-\ell(\theta)$ subject to $\Delta(\theta) \leq \delta$ or equivalently by minimizing the penalized negative log-likelihood $-\ell(\theta) + \lambda n\Delta(\theta)$ for $\lambda \geq 0$. We use this penalized negative log-likelihood to define the fair Gibbs posterior. That is, we use

$$\nu_n(\theta; \lambda) \propto \exp\big(\ell(\theta) - \lambda n\Delta(\theta)\big)\pi(\theta) \tag{5}$$

as variational distributions and find $\lambda$ that maximizes the ELBO subject to the average $\mathrm{DP}_\delta$ constraint. Since $\lambda$ is a scalar, a grid search could be used to estimate it. See Appendix A.2 for the detailed calculation of ELBO.

### 4.1 MATCHED DEVIATION

In this section, we propose a deviation so-called *matched deviation*. A key advantage of the matched deviation is that the corresponding Gibbs posterior does not require adversarial learning.

A function $\mathbf{T} : \mathcal{X}_1 \to \mathcal{X}_0$ is called a matching function if $\mathbf{T}_\#\mathbb{P}_1 = \mathbb{P}_0$, where $\mathbf{T}_\#\mathbb{P}_1$ is the push-forward measure of $\mathbb{P}_1$ induced by $\mathbf{T}$. For a given matching function $\mathbf{T}$, the matched deviation of $f$ is defined as

$$\Delta_{\mathrm{M}}(\theta, \mathbf{T}) := \mathbb{E}_{X_1 \sim \mathbb{P}_1}(\|f_\theta(X_1, s=1) - f_\theta(\mathbf{T}(X_1), s=0)\|^2), \tag{6}$$

where $\|\cdot\|$ is the Euclidean norm. Note that when $\mathbb{P}_0$ and $\mathbb{P}_1$ are empirical distributions, the matching function $\mathbf{T}$ reduces to a mapping between individual samples.

A notable property of the matched deviation is that it is an upper bound of the Wasserstein norm regardless of the choice of matching function $\mathbf{T}$, and there exists a matching function $\mathbf{T}$ making the matched deviation upper bounded when the total variation norm is bounded. These are formally stated in the following Theorems 4.1 and 4.2, whose proofs are deferred to Appendix A.3.

**Theorem 4.1** ($\Delta_{\mathrm{M}} \Rightarrow \Delta_{\mathrm{W}}$). *For all $\mathbf{T} : \mathcal{X}_1 \to \mathcal{X}_0$, if $\Delta_{\mathrm{M}}(\theta, \mathbf{T}) \leq \delta$ holds for $\delta \geq 0$, then we have $\Delta_{\mathrm{W}}(\theta) \leq \delta$.*

**Theorem 4.2** ($\Delta_{\mathrm{TV}} \Rightarrow \Delta_{\mathrm{M}}$). *If $\Delta_{\mathrm{TV}}(\theta) \leq \delta$ holds for some $\delta \in [0, 1]$, then there exists a matching function $\mathbf{T} : \mathcal{X}_1 \to \mathcal{X}_0$ satisfying $\Delta_{\mathrm{M}}(\theta, \mathbf{T}) \leq 2c\delta$ for some $c > 0$ not depending on $\mathbf{T}$.*

### 4.2 MATCHED GIBBS POSTERIOR

Motivated by the above theorems, we consider the following penalized log-likelihood $\ell(\theta) - \lambda n\Delta_{\mathrm{M}}(\theta, \mathbf{T})$. In particular, Theorem 4.1 implies that we can control the group fairness of the minimizer of the above penalized log-likelihood by controlling $\lambda$ accordingly. On the other hand, Theorem 4.2 implies that any group-fair prediction model $f$ in terms of the total variation norm has a matching function $\mathbf{T}$ with small $\Delta_{\mathrm{M}}(f, \mathbf{T})$. These interesting results suggest the following Gibbs posterior:

$$\nu_{\mathrm{M}}(f, \mathbf{T}|\lambda) \propto \exp\big(\ell(f) - \lambda n\Delta_{\mathrm{M}}(f, \mathbf{T})\big)\pi(f)\pi(\mathbf{T}), \tag{7}$$

which we call matched Gibbs posterior. A notable feature is that, we treat $\mathbf{T}$ as well $f$ as the parameter to be inferred. By doing so, we can avoid an adversarial learning even though additional computation is required for treating $\mathbf{T}$. For generating samples from $\nu_{\mathrm{M}}(f, \mathbf{T}|\lambda)$, it is natural to use the Gibbs sampler algorithm where $f \sim \nu_{\mathrm{M}}(f|\mathbf{T}, \lambda)$ and $\mathbf{T} \sim \nu_{\mathrm{M}}(\mathbf{T}|f, \lambda)$ are repeated until convergence. Note that generating samples using a Metropolis-Hastings algorithm is computationally much easier than the optimization for the adversarial learning.

An additional advantage of matched Gibbs posterior is that $\nu_{\mathrm{M}}(f|\mathbf{T}, \lambda)$ can have a simple form for certain problems. For example, consider a regression problem $Y_i = f(X_i) + \epsilon_i$, where $\epsilon_i \sim N(0, \sigma^2)$ are independent noises. When $f$ is a Gaussian process a priori and $\sigma^2$ is known, then $\nu_{\mathrm{M}}(f|\mathbf{T}, \lambda)$ also becomes a Gaussian process and hence a sample can be generated easily (see Appendix A.4 for the detailed derivation). When either the likelihood is not Gaussian or $f$ is not Gaussian a priori, a sample from $\nu_{\mathrm{M}}(f|\mathbf{T}, \lambda)$ can be generated by use of well known techniques such as Hamiltonian Monte-Carlo (HMC) (Neal et al., 2011). Generating $\mathbf{T}$ from its conditional posterior needs a specially designed algorithm, which is the topic for the next subsection.

*Remark* 4.3. Instead of learning $\mathbf{T}$ together with $f$, we may fix it at a certain matching function to save computation needed to generate $\mathbf{T}$. An example is the optimal transport between $\{X_i : S_i = 0\}$ and $\{X_i : S_i = 1\}$ if there is a reasonable metric on $\mathcal{X}$. Kim et al. (2025a) showed that the minimizer of the penalized log-likelihood with the optimal transport for $\mathbf{T}$ has many desirable properties such as improvement on individual fairness metrics without hampering group fairness. Our numerical studies in Appendix C.6 implies that such desirable properties are still valid for matched Gibbs posterior. Note, however, that this approach is not applicable when $\mathcal{X}$ is not a metric space (e.g., categorical data, text data) and yields suboptimal results.

## 4.3 MCMC ALGORITHM

For the posterior inference, we consider an MCMC (Markov Chain Monte-Carlo) algorithm. Simply, we can utilize some well-known sampling methods as HMC, to yield samples $\theta$ from $\nu(\theta; \lambda)$. However, for matched Gibbs posterior, we need an additional step to generate $\mathbf{T}$ from its conditional posterior. Here, we adopt a Gibbs sampler to iteratively sample $\theta \sim p(\theta|\mathbf{T}, \mathcal{D}_n)$ and $\mathbf{T} \sim p(\mathbf{T}|\theta, \mathcal{D}_n)$.

Sampling $\theta \sim p(\theta|\mathbf{T}, \mathcal{D}_n)$ is equivalent with $\theta^{(t)} \sim \nu_n(\theta; \lambda)$ for the given $\mathbf{T}$. We utilize HMC for this step. For sampling of $\mathbf{T}$, we consider a Metropolis-Hastings (MH) algorithm with regard $\mathbf{T}$.

To infer $\mathbf{T}$, we have to specify the prior. Motivated by Volkovs & Zemel (2012), we consider $e(\mathbf{T})$, the energy of $\mathbf{T}$ which is defined as:

$$e(\mathbf{T}) = e(\mathbf{T}; \tau) := \exp\left( -\sum_{i=1}^{n_1} d\big(X_i^{(0)}, \mathbf{T}(X_i^{(1)})\big)/n_0\tau \right) \tag{8}$$

with a pre-specified distance measure $d$ and temperature $\tau > 0$. We let $\pi(\mathbf{T}) \propto e(\mathbf{T})$.

For the proposal $\mathbf{T} \to \mathbf{T}'$ in the MH algorithm, we randomly select $k$ indices $i_1, \ldots, i_k$ from $[n_1]$, and define as:

$$\mathbf{T}'(j) := \begin{cases} \mathbf{T}(j) & \text{for } j \notin [n_1] \setminus \{i_1, \ldots, i_k\}, \\ \mathbf{T}(i_{\Pi_k(l)}) & \text{for } j = i_l, \end{cases} \tag{9}$$

where $\Pi_k$ is a random permutation of $[k]$. Here, $k$ is pre-specified hyperparameter. The acceptance probability can be easily calculated by the posterior ratio, since the proposals are totally random. See Appendix C.1 for a visualization of the proposal $\mathbf{T}'$ in Fig. 4, and the acceptance probability.

## 4.4 SELECTION OF $\lambda$

Since $\lambda$ is a real-valued constant, we find the optimal $\lambda$ by a grid-search. As usual, we first choose the set $\Lambda_{\mathrm{cand}}$ of finitely many candidates for $\lambda$. Then, for each $\lambda \in \Lambda_{\mathrm{cand}}$, we generate posterior samples of $\theta$ and $\mathbf{T}$ and calculate the ELBO and average DP. Finally, we choose $\lambda$ which maximizes the ELBO among those whose average DP is less than $\delta$. See Appendix B.2 for $\lambda$ values that we use to provide the results in our numerical experiments.

## 5 EXPERIMENTS

We conduct numerical experiments on various benchmark datasets[3]. For the prediction models, we use DNNs since DNNs are vulnerable to overfitting and thus uncertainty quantification is necessary. See Appendix B.2 for more details of the experimental setting. See Appendix C, D for omitted results and ablation studies, respectively.

### 5.1 SETTINGS

**Datasets**  We analyze the following five benchmark datasets that are commonly used for group-fair classification: ADULT, DUTCH, CRIME, CELEBA, and CIVIL. Brief explanations for each dataset are given below, with details in Appendix B.1.

Tabular datasets:

    (i) ADULT: Adult dataset (Becker & Kohavi, 1996) has a label whether income from an individual is larger than $50K/yr based on census data. We consider 'gender' as a sensitive variable.

   (ii) DUTCH: Dutch census dataset (Van der Laan, 2000; Quy et al., 2022) has a label whether a person's occupation can be categorized as high-level or low-level. We consider 'gender' as a sensitive variable.

  (iii) CRIME: Communities & Crime dataset (Redmond & Baveja, 2002) includes a label of the number of violent crimes per 100,000 population, based on socio-economic, law enforcement, and crime data from communities. We binarize 'number of violent crimes' with its median and consider it as a label. We also binarize 'percentage of population that is African-Ameraicn' with its median and consider it as a sensitive variable.

Image dataset:

  (iv) CELEBA: CelebAMask-HQ dataset (Lee et al., 2020) is a face image dataset, where facial attributes are annotated with binary labels. We consider 'Male' as a sensitive variable, and predict 'Attractive' as a target variable.

Text dataset:

  (v) CIVIL: CivilComments-Wilds dataset (Borkan et al., 2019; Koh et al., 2021) is a text classification data to identify whether the comment is toxic or not. We consider two race groups: 'black' and 'asian', since those groups show the largest gap in the proportion of toxic comments.

**Learning algorithms**  We consider 3 fair-Bayesian methods: mean-field Gaussian with MMD, Gibbs posterior with MMD, and matched Gibbs posterior. We also consider 3 most popular state-of-the-art algorithms for demographic parity mitigation, GapReg (Donini et al., 2018; Chuang & Mroueh, 2021), Reduction (Agarwal et al., 2018a) and Adv (Zhang et al., 2018), as deterministic baseline methods. These 6 methods are annotated in the figures and tables as: *variational_mmd*, *gibbs_mmd*, *gibbs_matched*, *gapreg*, *reduction* and *adv*. Note that the computation of MMD cannot be done by mini-batches, hence we consider full-batch for the 2 MMD based methods. Due to computational burden in calculating MMD which requires $O(n^2)$ operations, *variational_mmd* and *gibbs_mmd* are only applicable when the dataset size is moderate, hence we use them only for CRIME. Also, we performed *adv* only for tabular datasets, due to its numerical instability. See Appendix B.2 for more details about these 6 learning algorithms.

**Evaluation metrics**  We measure the prediction performance of the posteriors or estimators of each learning algorithm on test data in terms of prediction utility as well as uncertainty quantification. For prediction utility, we consider the classification accuracy of the Bayes estimator (`Acc`). For uncertainty quantification, we consider the negative log-likelihood (`Nll`), Brier score (`brier`) (Brier, 1950), and expected calibration error (`Ece`) (Guo et al., 2017) of the predictive distribution. Higher values of `Acc` mean better prediction performances while lower values of `Nll`, `brier` and `Ece` mean better uncertainty quantification. For group fairness measure, we use $\Delta_{\mathrm{W}}^{1/2} = \mathcal{W}_2(\mathbb{P}_{f,0}, \mathbb{P}_{f,1})$.

---

[3]Our code is available at `https://github.com/JihuLee/MatchedGibbs`

We take a square root to $\Delta_W$, to preserve the original scale of Wasserstein distance $\mathcal{W}_2$. See Appendix B.2 for the definitions of the performance measures.

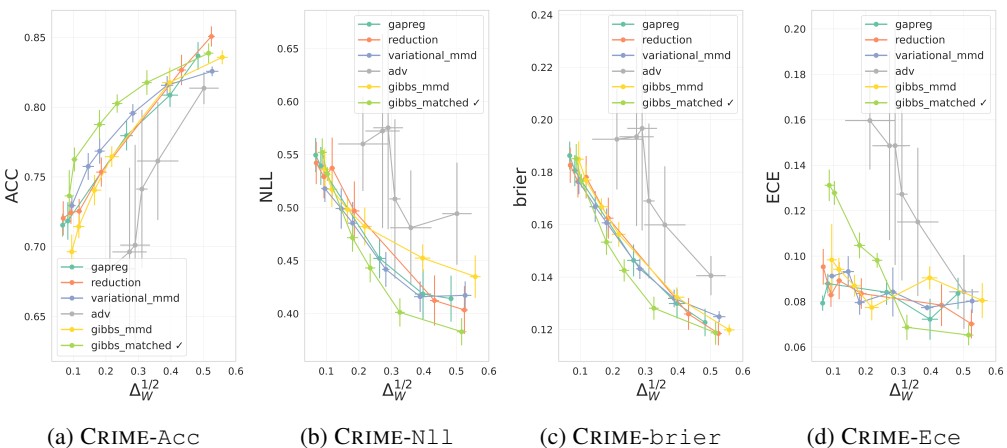

(a) CRIME-Acc  (b) CRIME-Nll  (c) CRIME-brier  (d) CRIME-Ece

Figure 1: **Tabular classification.** Pareto-front lines between level of $\Delta_W^{1/2}$ (on the $x$-axis) and the predictive performance (on the $y$-axis), on CRIME.

## 5.2 RESULTS

We plot the Pareto-front lines between the level of group fairness measured by $\Delta_W^{1/2}$ and the evaluation metrics for predictive performance. We use the averages of the fairness levels and metrics for predictive performance on 5 repeated experiments, along with their error bars. We can compare the efficiency of each trade-off using them.

Fig. 1 provides the results on CRIME. Results for other tabular datasets (ADULT and DUTCH) are provided in Appendix C.4. We can clearly observe that the trade-off of both utility (Acc) and uncertainty (Nll, brier) is superior to that of other algorithms, in all 3 datasets. An interesting point is that we can observe that *gibbs_mmd* and *variational_mmd* does not perform well in terms of Acc, Nll and brier. Hence, we conclude that even in the sole practical case of moderate-size dataset, both methods are not feasible competitors to *gibbs_matched*. For Ece, the results of *gibbs_matched* on ADULT show the best trade-offs. Results on CRIME and DUTCH, show the best performance when the group fairness level is not too strict, and becomes less favorable when the group fairness level becomes strict. In this regard, we argue that a low Ece alone does not guarantee a good model. For example, a model that assigns a constant confidence of 0.5 to every test sample would yield an Ece close to zero, yet its accuracy would be poor. This can be supported by the result of *adv* on DUTCH in Fig. 7, placed in Appendix C.4, where its performance on (Acc, Nll, brier) are not favorable than others, while maintaining relatively good trade-off in Ece.

Fig. 2 provides the result on CELEBA. We can observe that *gibbs_matched* shows the best performance in terms of Acc with large margins. A similar behavior is observed in the results on CIVIL in Fig. 3. Also, for both datasets, the uncertainty measures of *gibbs_matched* consistently achieve the lowest (best) compared to deterministic baselines, while maintaining high Acc trade-off. This is because Bayesian inference generally enhances uncertainty measures compared to deterministic algorithms (Gal et al., 2016; Lakshminarayanan et al., 2017). Note that the increase in Nll of *gapreg* and *reduction* as the fairness level increases, is due to their overfitting, even with the best validation selection.

Another benefit of the use of matched Gibbs posterior is that, it also improves individual fairness metrics, similar to the result from Kim et al. (2025a). This is because the matched deviation directly minimizes the gap between the output from different individuals. We empirically validate that matched Gibbs posterior shows superior performance on Con (consistency score (Zemel et al., 2013; Yurochkin et al., 2020; Yurochkin & Sun, 2021)) that is a metric for individual fairness, than that of other baselines. See Appendix C.6 for detailed results. We also investigate the convergence of the proposed MCMC algorithm for matched Gibbs posterior in Appendix C.7, through the traceplots

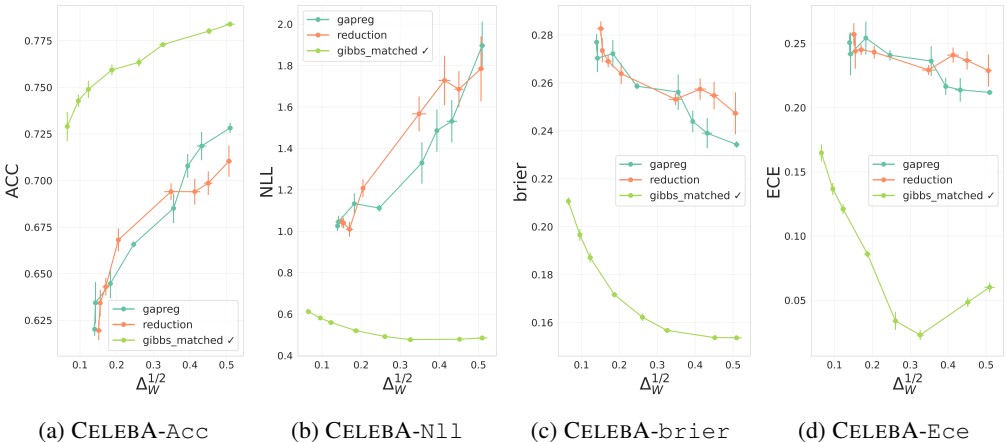

Figure 2: **Image classification.** Pareto-front lines between level of $\Delta_W^{1/2}$ (on the $x$-axis) and the predictive performance (on the $y$-axis). The trade-off of proposed matched Gibbs posterior is superior to that of other competitors.

related to $\theta$ and acceptance probability of $\mathbf{T}$. The results amply suggest that the proposed MCMC algorithm converges well.

In summary, from the experiments on various real-world datasets, we can conclude that matched Gibbs posterior does improve the utility-fairness trade-off and uncertainty-fairness trade-offs. Specifically, matched Gibbs posterior shows (i) substantially better trade-offs in (Acc, Nll and brier), and (ii) superior trade-offs in Ece, on ADULT, CELEBA and CIVIL. Also, matched Gibbs posterior (iii) exhibits some an additional benefit of improved individual fairness, and (iv) does converge well. Therefore, we conclude that matched Gibbs posterior is a favorable proxy distribution for the fair variational Bayesian inference.

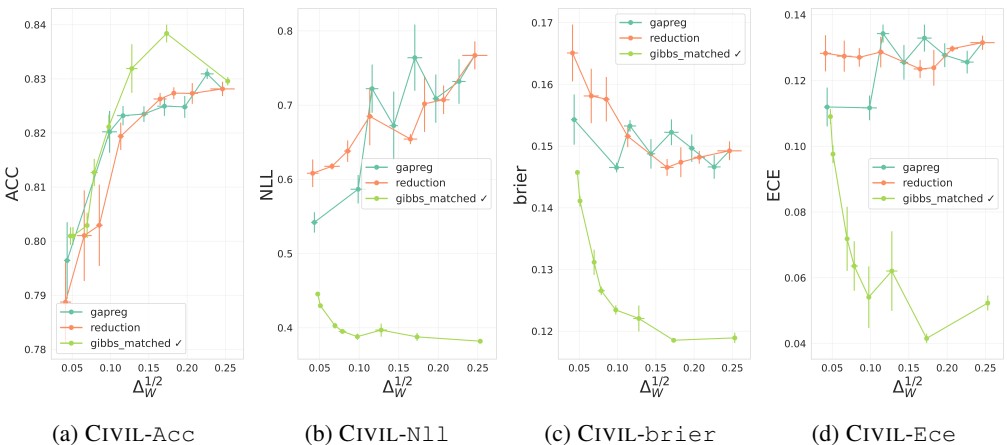

Figure 3: **Text classification.** Pareto-front lines between level of $\Delta_W^{1/2}$ (on the $x$-axis) and the predictive performance (on the $y$-axis). The trade-off of proposed matched Gibbs posterior is superior to that of other competitors.

## 5.3 ABLATION STUDIES

We perform several ablations studies, including (i) effects of $\tau$, (ii) effect of pretrain epochs, (iii) effect of the change of the direction of $\mathbf{T}$ by flipping the label of sensitive attributes, and (vi) effect of prior choice. See Appendix D for details and results. In each study, we still observe that matched Gibbs posterior maintains its superior trade-offs under varying conditions.

### 5.4 Additional results

We also provide additional results including (i) extension of matched Gibbs posterior to Equalized Odds (EO), (ii) extension to multinary sensitive attribute case, (iii) performance evaluated with Strong Demographic Parity (SDP) (Jiang et al., 2020a; Silvia et al., 2020), (iv) measurement on computational costs, and (v) additional experiment on ACSIncome dataset (Han et al., 2023). See Appendix E for details and results. These results provide additional support for the effectiveness of matched Gibbs posterior.

## 6 Concluding Remarks

We introduced a variational Bayesian inference framework for learning group-fair posteriors. Then, we proposed a novel proxy distribution named matched Gibbs posterior, based on the matched deviation that can effectively control the group fairness of the posteriors. We show that the matched deviation possesses several theoretical advantages and that matched Gibbs posterior achieves superior performance over other baseline methods along with some additional benefits such as individual fairness.

We mainly focus on binary sensitive attributes in this work. Extension of matched Gibbs posterior to multinary sensitive attributes could be done similarly to what (Kim et al., 2025b) has done in Section A.3. More detailed explanation for this extension is described in Appendix E.2. We will report related algorithms and results in a near future.

Study of theoretical properties of matched Gibbs posterior such as posterior consistency would be worth pursuing. In addition, Bayesian analysis for other group fairness aware tasks such as representation learning (Zemel et al., 2013; Madras et al., 2018; Kim et al., 2022) would be a promising future work.

### Ethics Statement

This paper studies group fairness in classification using only public datasets; no new human-subject data were collected. The fairness notions and performance measures are also widely used in recent works. We believe the framework helps prevent discrimination in classification supported by theoretical guarantees, and would not raise new critical societal concerns.

### Reproducibility Statement

We have made significant efforts to ensure the reproducibility of our findings in this study. For the theoretical results, we present complete proofs in Appendix. The source codes for implementing our proposed model and running conducted experiments are provided in the supplementary material. Detailed information for the hyperparameters, datasets, experimental setup are given in Appendix B.2.

### Acknowledgments

This work was partly supported by Institute of Information & communications Technology Planning & Evaluation (IITP) grant funded by the Korea government(MSIT) (No.RS-2022-II220184, Development and Study of AI Technologies to Inexpensively Conform to Evolving Policy on Ethics), the National Research Foundation of Korea(NRF) grant funded by the Korea government(MSIT)(No. 2022R1A5A7083908), the National Research Foundation of Korea(NRF) grant funded by the Korea government(MSIT) (RS-2025-00556079), and Institute of Information & communications Technology Planning & Evaluation (IITP) grant funded by the Korea government(MSIT) [NO.RS-2021-II211343, Artificial Intelligence Graduate School Program (Seoul National University)].

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

# APPENDIX

## A  THEORETICAL DETAILS

### A.1  DEFINITIONS OF DEVIANCES

Consider two random variables $x$ and $y$, which follows the probability distribution $\mathcal{P}$ and $\mathcal{Q}$, respectively. The formerly mentioned deviances of $\mathcal{P}$ and $\mathcal{Q}$ can be defined as follows.

The (2-)Wasserstein distance is defined as:

$$\mathcal{W}_2(\mathcal{P}, \mathcal{Q}) := \inf_{\gamma \in \Gamma(\mathcal{P}, \mathcal{Q})} \mathbb{E}_{(x,y) \sim \gamma} \|x - y\|_2. \tag{10}$$

Here, $\Gamma(\mathcal{P}, \mathcal{Q})$ is the set of joint probability distributions, whose marginals are $\mathcal{P}$ and $\mathcal{Q}$.

The total variation distance is defined as:

$$\mathrm{TV}(\mathcal{P}, \mathcal{Q}) := \sup_{A \in \mathcal{A}} |\mathcal{P}(A) - \mathcal{Q}(A)|. \tag{11}$$

Here, $\mathcal{A}$ is the set of all measurable subsets.

The Kolmogorov-Smirnov distance is defined as:

$$\mathrm{KS}(\mathcal{P}, \mathcal{Q}) := \sup_{t \in \mathbb{R}} |F_{\mathcal{P}}(t) - F_{\mathcal{Q}}(t)|. \tag{12}$$

Here, $F_{\mathcal{P}}(t) := \mathcal{P}(x \leq t)$ is the cumulative distribution of $\mathcal{P}$.

### A.2  CALCULATION OF ELBO

The evidence lower-bound (ELBO) is defined as:

$$\mathrm{ELBO}(\nu) := \mathbb{E}_{f \sim \nu}[\log \mathcal{L}(\mathcal{D}_n; f)] - D_{\mathrm{KL}}\big(\nu(f) \| \pi(f)\big). \tag{13}$$

In this section, we provide the calculation of ELBO when considering $\nu_n(\theta; \lambda)$. This calculation can be used to the selection of $\lambda$, under certain level of averaged DP. Note that the first term can be easily calculated using samples yielded from $\nu$, by Monte-Carlo approximation.

We can also calculate the ELBO in general cases. We have:

$$\nu_n(\theta; \lambda) = \frac{\pi(\theta) e^{-R_n(\theta; \lambda)}}{Z_\lambda}, \quad Z_\lambda = \int \pi(\theta) e^{-R_n(\theta; \lambda)} df = \mathbb{E}_{\theta \sim \pi}[e^{-R_n(\theta; \lambda)}], \tag{14}$$

where $R_n(\theta; \lambda) := -\ell(\theta) + \lambda n \Delta(\theta)$. Hence, we can simplify the KL term as follows:

$$D_{\mathrm{KL}}(\nu_n(\cdot; \lambda) \| \pi(\cdot)) = \mathbb{E}_{\theta \sim \nu_n(\cdot; \lambda)}[\log \nu_n(\theta; \lambda) - \log \pi(\theta)] = \mathbb{E}_{\theta \sim \nu_n(\cdot; \lambda)}[-R_n(\theta; \lambda)] - \log Z_\lambda. \tag{15}$$

Now, using samples $(\theta^{(t)})_{t=1}^T \sim \nu_n(\cdot; \lambda)$ and $(\tilde{\theta}^{(s)})_{s=1}^S \sim \pi(\cdot)$, we can calculate the KL term by Monte-Carlo approximation as follows:

$$D_{\mathrm{KL}}(\nu_n(\cdot; \lambda) \| \pi(\cdot)) \approx -\frac{1}{T} \sum_{t=1}^T R_n(\theta^{(t)}; \lambda) - \log \left( \frac{1}{S} \sum_{s=1}^S \exp\{-R_n(\tilde{\theta}^{(s)}; \lambda)\} \right). \tag{16}$$

For the numerical stability, one could calculate the last term by utilizing log-sum-exp.

### A.3  PROOFS OF THE BOUNDS RELATED TO THE MATCHED DEVIANCE

*Proof of Theorem 4.1.* Recall the definitions of the deviances that

$$\Delta_{\mathrm{W}}(\theta) := W_2^2(P_{f_{\theta,1}}, P_{f_{\theta,0}}), \quad \Delta_{\mathrm{M}}(\theta, \mathbf{T}) := \mathbb{E}\left\| f_\theta(X, 1) - f_\theta(\mathbf{T}(X), 0) \right\|_2^2 \tag{17}$$

for a given matching $\mathbf{T} : \mathcal{X}_1 \to \mathcal{X}_0$ satisfying $\mathbf{T}_\# \mathbb{P}_1 = \mathbb{P}_0$.

For a simplicity, let $\mu_s = \mathrm{Law}(P_{f_\theta,s})$ for $s \in \{0,1\}$. Hence, by the Kantorovich formulation of the Wasserstein distance $W_2$,

$$W_2^2(P_{f_\theta,1}, P_{f_\theta,0}) \le \mathbb{E}\|f_\theta(X,1) - f_\theta(\mathbf{T}(X),0)\|_2^2 \tag{18}$$

for any $\mathbf{T}$. Taking the infimum over $\mathbf{T}$ on the right-hand side yields:

$$W_2^2(\mu_1, \mu_0) \le \inf_{\mathbf{T}} \mathbb{E}\|f_\theta(X,1) - f_\theta(\mathbf{T}(X),0)\|_2^2. \tag{19}$$

Therefore,

$$\Delta_{\mathrm{W}}(\theta) \le \inf_{\mathbf{T}} \Delta_{\mathrm{M}}(\theta, \mathbf{T}). \tag{20}$$

In particular, if there exists $\mathbf{T}$ with $\Delta_{\mathrm{M}}(\theta, \mathbf{T}) \le \delta$, then $\Delta_{\mathrm{W}}(\theta) \le \delta$.

$\square$

*Proof of Theorem 4.2.* Fix $\eta > 0$ and partition the cube $[0,1]^c$ into a finite family of disjoint cubes $\{B_k\}_{k=1}^m$ of length $\eta$ for each side. Then, we have $\mathrm{diam}(B_k) = \sqrt{c}\eta$ and $\mathrm{diam}([0,1]^c) = \sqrt{c}$.

Let $A_{1,k} := f_\theta^{-1}(B_k) \cap \{S = 1\} \subset \mathcal{X}_1$ and $A_{0,k} := f_\theta^{-1}(B_k) \cap \{S = 0\} \subset \mathcal{X}_0$. Write $p_k := \mathbb{P}_1(A_{1,k}) = P_{f_\theta,1}(B_k)$ and $q_k := \mathbb{P}_0(A_{0,k}) = P_{f_\theta,0}(B_k)$. Define $r_k := (p_k - q_k)_+$, $s_k := (q_k - p_k)_+$, and $m_k := \min\{p_k, q_k\} = p_k - r_k = q_k - s_k$. Let $R := \sum_{k=1}^m r_k = \sum_{k=1}^m s_k \le \frac{1}{2}\sum_{k=1}^m |p_k - q_k| \le \mathrm{TV}(P_{f_\theta,1}, P_{f_\theta,0}) \le \delta$. Here, note that $\sum r_k = \sum s_k$ since $\sum p_k = \sum q_k = 1$. Note also that $\sum_{k=1}^m m_k = \sum_{k=1}^m (p_k - r_k) = 1 - \sum_{k=1}^m r_k = 1 - R$.

Then, for each $k$ there exist measurable subsets $A_{1,k}^{\mathrm{mat}} \subseteq A_{1,k}$ and $A_{0,k}^{\mathrm{mat}} \subseteq A_{0,k}$ with $\mathbb{P}_1(A_{1,k}^{\mathrm{mat}}) = \mathbb{P}_0(A_{0,k}^{\mathrm{mat}}) = m_k$, and a bijection $\mathbf{T}_k^{(1)} : A_{1,k}^{\mathrm{mat}} \to A_{0,k}^{\mathrm{mat}}$ satisfying $\mathbb{P}_0(B) = \mathbb{P}_1\big((\mathbf{T}_k^{(1)})^{-1}(B)\big)$ for all $B \subseteq A_{0,k}^{\mathrm{mat}}$.

Let $A_1^{\mathrm{rem}} := \mathcal{X}_1 \setminus \bigcup_k A_{1,k}^{\mathrm{mat}}$ and $A_0^{\mathrm{rem}} := \mathcal{X}_0 \setminus \bigcup_k A_{0,k}^{\mathrm{mat}}$ be the residual domains. By construction, $\mathbb{P}_1(A_1^{\mathrm{rem}}) = \mathbb{P}_0(A_0^{\mathrm{rem}}) = R \le \delta$. Then, there exists a bijection $\mathbf{T}^{(3)} : A_1^{\mathrm{rem}} \to A_0^{\mathrm{rem}}$.

Define $\mathbf{T} : \mathcal{X}_1 \to \mathcal{X}_0$ piecewise by

$$\mathbf{T}(x) := \begin{cases} \mathbf{T}_k^{(1)}(x), & x \in A_{1,k}^{\mathrm{mat}} \text{ for } k \in [m], \\ \mathbf{T}^{(3)}(x), & x \in A_1^{\mathrm{rem}}. \end{cases} \tag{21}$$

$\mathbf{T}$ is well-defined as a countable union of maps on disjoint domains. Moreover, for any $B \subseteq \mathcal{X}_0$ that is a finite union of pieces from $\{A_{0,k}^{\mathrm{mat}}\}_k$ and $A_0^{\mathrm{rem}}$, we have $\mathbb{P}_0(B) = \mathbb{P}_1(\mathbf{T}^{-1}(B))$ and $\mathbf{T}\#\mathbb{P}_1 = \mathbb{P}_0$.

If $x \in A_{1,k}^{\mathrm{mat}}$, then $f_\theta(x,1)$ and $f_\theta(\mathbf{T}(x),0)$ both lie in the same cube $B_k$, so $\|f_\theta(x,1) - f_\theta(\mathbf{T}(x),0)\|_2^2 \le \big(\sqrt{c}\eta\big)^2 = c\eta^2$. If $x \in A_1^{\mathrm{rem}}$, we use a trivial bound that $\|\cdot\|_2^2 \le \mathrm{diam}([0,1]^c)^2 = c$, so that we have $\Delta_{\mathrm{M}}(\theta, \mathbf{T}) = \mathbb{E}_{X_1}\|f_\theta(X_1,1) - f_\theta(\mathbf{T}(X_1),0)\|_2^2 \le (1-R)c\eta^2 + Rc \le c\big((1-\delta)\eta^2 + \delta\big)$, since $R \le \delta$.

Taking $\eta := \sqrt{\delta} \in [0,1]$, we have $\Delta_{\mathrm{M}}(\theta, \mathbf{T}) \le c\big((1-\delta)\delta + \delta\big) = c(2\delta - \delta^2) \le 2c\delta$, which concludes the proof. (Note that the proof also holds for regression problems, when the output $y$ is scaled into $[0,1]$.) $\square$

## A.4 MATCHED GIBBS POSTERIOR FOR GAUSSIAN REGRESSION PROBLEMS

In Section 4.2, we noted that matched Gibbs posterior can be computed easily for a certain problems. In this section, we provide the explicit calculation for Gaussian regression problems.

Consider a simple regression problem of:

$$Y_i = f(X_i, S_i) + \epsilon_i, \ \epsilon_i \sim N(0, \sigma^2). \tag{22}$$

Assume that $f \sim \mathcal{GP}(0, k(\cdot, \cdot))$, where $\mathcal{GP}$ denotes a Gaussian process. Let $y = (y_1, \ldots, y_n)^\top$ and $K$ be a matrix with $K_{ij} = k((x_i, s_i), (x_j, s_j))$.

Then the log-likelihood is given as:

$$\ell(f) = \log \mathcal{L}(f; \mathcal{D}_n) = -\frac{1}{2\sigma^2}||y - \boldsymbol{f}||_2^2 - \frac{n}{2}\log(2\pi\sigma^2), \tag{23}$$

where $\boldsymbol{f} = (f(x_1), \ldots, f(x_n))^\top$.

The constraint is calculated as:

$$\begin{aligned} n\Delta_{\mathrm{M}}(f, \mathbf{T}) &= ||(I - P_{\mathbf{T}})\, S_1 \boldsymbol{f}||_2^2 \\ &= \boldsymbol{f}^\top C_{\mathbf{T}} \boldsymbol{f}, \end{aligned} \tag{24}$$

Here, $S_1 \in \mathbb{R}^{m \times n}$ selects the coordinates with $S_i = 1$ and $P_T$ is the permutation matrix induced by the matching map $\mathbf{T}$ on those coordinates. $C_{\mathbf{T}}$ is defined as $S_1^\top (I - P_{\mathbf{T}})^\top (I - P_{\mathbf{T}}) S_1$.

Hence, the matched Gibbs posterior is simplified as:

$$\begin{aligned} \nu_{\mathrm{M}}(\boldsymbol{f}, \mathbf{T}; \lambda) &\propto \exp(\ell(f) - \lambda n \Delta_{\mathrm{M}}(f, \mathbf{T}))\pi(f)\pi(\mathbf{T}) \\ &\propto \exp\left(-\tfrac{1}{2}\boldsymbol{f}^\top \left(K^{-1} + \sigma^{-2}\mathbb{I}_n + 2\lambda C_{\mathbf{T}}\right)\boldsymbol{f} + \sigma^{-2}y^\top \boldsymbol{f}\right)\pi(\mathbf{T}) \\ &\propto \exp\left(-\tfrac{1}{2}\boldsymbol{f}^\top \Lambda_{\mathbf{T}} \boldsymbol{f} + \sigma^{-2}y^\top \boldsymbol{f}\right)\pi(\mathbf{T}). \end{aligned} \tag{25}$$

Here, $\Lambda_{\mathbf{T}} := K^{-1} + \sigma^{-2}\mathbb{I}_n + 2\lambda C_{\mathbf{T}}$. That is, the conditional posterior of $f$ given $\mathbf{T}$ and $\sigma^2$ is again a Gaussian process.

## B    EXPERIMENTAL DETAILS

### B.1    DATASETS

We first explain about benchmark datasets that we used in experiments.

(i) ADULT: Adult dataset (Becker & Kohavi, 1996) has a label whether income from an individual is larger than $50K/yr based on census data. We consider sensitive variable as gender. Preprocessing procedure follows from (Yurochkin et al., 2020), removing some features in the dataset, and one-hot encode the discrete and quantized continuous variables.

(ii) DUTCH: Dutch census dataset (Van der Laan, 2000; Quy et al., 2022) has a label whether a person's occupation can be categorized as high-level or low-level. We consider sensitive variable as gender. We choose categorical features (`age`, `household_position`, `household_size`, `citizenship`, `country_birth`, `edu_level`, `economic_status`, `cur_eco_activity`, `marital_status`), and expanded them by one-hot encoding.

(iii) CRIME: Communities & Crime dataset (Redmond & Baveja, 2002) has a label of the number of violent crimes per 100,000 population, based on socio-economic, law enforcement, and crime data about communities. The label - number of violent crimes - is divided into binary label by its median, and the sensitive variable - percentage of population that is african american - is also divided into binary sensitive groups by its median. We dropped (`state`, `country`, `community`, `fold`, `communityname`).

(iv) CELEBA: CelebAMask-HQ is a face image dataset with 30,000 images, annotated with 19 binary facial attributes (e.g., `eyeglasses`, `smiling`, `wavy hair`,...). We consider the attribute `attractive` as a target label and `male` as a sensitive variable (Female = 1, Male = 0). We utilize the pretrained ResNet18 to yield the image embeddings of dimension 512.

(v) CIVIL: CivilComments-Wilds (Koh et al., 2021) is a processed version of the original CivilComments dataset (Borkan et al., 2019). We consider two sensitive variables from `race` - 'black' and 'asian', since the proportion of toxic comments exhibits the largest gap between them. We utilize the pretrained Roberta (Liu et al., 2019) to yield a sentence embedding of dimension 768.

We split datasets into two partitions with a proportion $8 : 2$, corresponding to train and test datasets, respectively, except CRIME. For CRIME, we randomly pick one from predefined train / test split candidates. We use a ReLU network with 2 hidden layers of width 200 for tabular datasets. For image and text datasets, we use a ReLU network with 2 hidden layers of width 512. Table 1 provides the basic descriptions of the 5 datasets.

Table 1: Summary statistics for each dataset.

| Dataset | Features ($d$) | Size | $Y = 1$ | $S = 1$ | train samples | test samples |
|---------|----------------|------|---------|---------|---------------|--------------|
| ADULT | 120 | 45,222 | 11,208 | 30,527 | 36,177 | 9,045 |
| DUTCH | 58 | 60,420 | 28,763 | 30,147 | 48,336 | 12,084 |
| CRIME | 121 | 1,994 | 1,042 | 1,038 | 1,794 | 200 |
| CELEBA | 512 | 30,000 | 17,218 | 18,943 | 24,000 | 6,000 |
| CIVIL | 768 | 33,613 | 7,714 | 12,059 | 26,890 | 6,723 |

### B.2    TRAINING DETAILS

**Learning algorithms**    For deterministic baselines (*gapreg*, *reduction* and *adv*), we randomly divided the train set in to two, with proportions 0.8 and 0.2 for train set and validation set, respectively. Then we save the model with via the performance on the validation set.

**Evaluation metrics**    As mentioned, we provide 1 utility measure - `Acc` and 3 uncertainty quantification measures - `Nll`, `Ece`, and `Brier`. As mentioned in the main body, `Acc` is an accuracy of the Bayes estimator. For uncertainty quantification measure, we measure the performance of the predictive distribution. Specifically for binary classification, we can calculate the predictive probability of a test data $(x_i, y_i, s_i)$ as $\hat{p}_i = p(y_i = 1 | x_i, s_i, \mathcal{D}_n) \approx \frac{1}{T} \sum_{t=1}^{T} f_{\theta^{(t)}}(x_i, s_i)$. Considering

binary classification tasks, $\mathtt{Nll}$ can be calculated by the binary cross-entropy with $\{\hat{p}_i\}_{i=1}^n$. $\mathtt{Ece}$ is a measure for calibration scores (Guo et al., 2017), where higher score implies better calibration. The formal definition is as follows:

$$\text{ECE} = \sum_{m=1}^{M} \frac{|B_m|}{n} \left| \text{acc}(B_m) - \text{conf}(B_m) \right|. \tag{26}$$

Here, $B_m$'s are $M$ bins of predictions, with sizes $|B_m|$. The accuracy and the confidence of the bins are defined as follows: $\text{acc}(B_m) := \frac{1}{|B_m|} \sum_{i \in B_m} \mathbb{I}(\hat{y}_i = y_i)$, $\text{conf}(B_m) := \frac{1}{|B_m|} \sum_{i \in B_m} \hat{p}_i$. $\mathtt{Brier}$ is a strictly proper scoring rule for measuring the accuracy of probabilistic predictions (Brier, 1950), commonly used in binary classification tasks. The formal definition on a function is as follows:

$$\text{Brier-score} := \frac{1}{n} \sum_{i=1}^{n} (\hat{p}_i - y_i)^2. \tag{27}$$

**Other details** We use a python package $\mathtt{hamiltorch}$[4] (Cobb et al., 2019) to perform HMC. The followings are the detailed values for several experimental settings.

For *gapreg*, *reduction*, *adv* and *variational_mmd*, we trained them with batch size of $1,024$ for 100 epochs, with learning rate $0.001$ and Adam optimizer. For *variational_mmd*, we used 5 samples for train and evaluation, where the prediction probabilities from the samples are averaged and used in the performance calculation. For Gibbs posteriors, we also used 5 samples where the prior of parameters are set to the standard Gaussian distributions. Following hyperparameters are organized in the order of (CRIME, ADULT, DUTCH, CELEBA and CIVIL). For MCMC chains, we utilized (5,10,10,10,500) burn-in epochs with 10 thinning intervals, respectively, and used HMC with step size of $0.001$. To give an efficient initial value, we utilized pretrained DNN models as initial parameter values, with pretrain epochs of $(50, 30, 10, 20, 70)$, respectively. For the permutation size of $\Pi_k$ (larger $k$ implies faster mixing), we use $(10, 5, 10, 10, 10)$ for $k$, respectively. For faster convergence, we initialized $\mathbf{T}$ with based on the optimal transport theory (Villani, 2008), along with the former insights of Kim et al. (2025a). Specifically, we utilized the joint OT map from Kim et al. (2025a) where the computation of a cost matrix is as follows: $\mathbf{C}^\gamma := [c_{i,j}^\gamma] \in \mathbb{R}^{n_1 \times n_0}$ where $c_{i,j}^\gamma = \|X_i^{(1)} - X_j^{(0)}\| + \gamma |Y_i^{(0)} - Y_j^{(1)}|$ with some constant $\gamma > 0$. Here, $(X_i^{(s)}, Y_i^{(s)})$ are observations from $\mathcal{X}^s$, and $n_s$ are the number of observations for each. Sufficiently large $\gamma$ suppress to match individuals that are differently labeled. We utilized $\mathtt{POT}$[5] package to solve an optimal transport. The values of $\lambda$ for matched Gibbs posterior that we used are provided in Table 2.

All our experiments are conducted on an Intel(R) Xeon(R) Silver 4410Y CPU with 128GB RAM and NVIDIA GeForce RTX 4090 GPUs.

Table 2: $\lambda$ values that are used in the main experiments.

| Dataset | $\lambda$ |
|---|---|
| CRIME | $[0.0, 1.0, 2.0, 3.0, 9.0, 10.0]$ |
| ADULT | $[0.0, 0.5, 2.0, 3.0, 3.5, 5.0, 6.0]$ |
| DUTCH | $[0.0, 1.0, 2.0, 5.0, 8.0, 9.0, 10.0]$ |
| CELEBA | $[0.0, 0.5, 1.0, 1.5, 2.5, 4.5, 5.5, 10.0]$ |
| CIVIL | $[0.0, 1.0, 2.5, 5.0, 5.5, 7.0, 9.5, 10.0]$ |

---

[4]Hamiltorch: `https://github.com/AdamCobb/hamiltorch`
[5]POT: `https://pythonot.github.io/`

## C OMITTED RESULTS AND DETAILS

In this section, we provide additional details and results omitted due to the shortage of space, to support the results in the main manuscript.

### C.1 DETAILS OF MATCHED GIBBS POSTERIOR

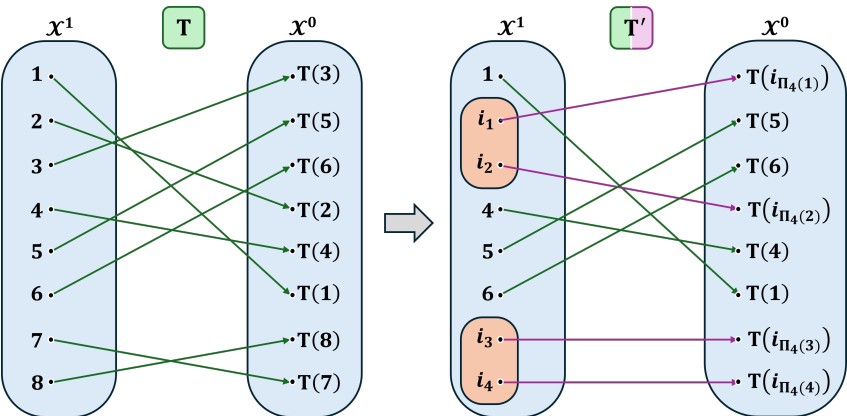

Figure 4: A visualization of the proposal construction of $\mathbf{T} \rightarrow \mathbf{T}'$.

The acceptance probability of a proposal $\mathbf{T}'$ can be calculated as:

$$\alpha(\mathbf{T}') = \min\{1, \alpha'(\mathbf{T}')\}. \tag{28}$$

Denote $\mathcal{L}$ as the likelihood function. Here, $\alpha'$ is defined as:

$$
\begin{aligned}
\alpha'(\mathbf{T}') &= \frac{p(\mathbf{T}'|\theta, \mathcal{D}_n)q((\mathbf{T}' \rightarrow \mathbf{T}))}{p(\mathbf{T}|\theta, \mathcal{D}_n)q(\mathbf{T} \rightarrow \mathbf{T}')} \\
&= \frac{p(\mathbf{T}')\mathcal{L}(\mathcal{D}_n; \theta, \mathbf{T}')q(\mathbf{T}' \rightarrow \mathbf{T})}{p(\mathbf{T})\mathcal{L}(\mathcal{D}_n; \theta, \mathbf{T})q(\mathbf{T} \rightarrow \mathbf{T}')} \\
&= \frac{e(\mathbf{T}')\mathcal{L}(\mathcal{D}_n; \theta, \mathbf{T}')}{e(\mathbf{T})\mathcal{L}(\mathcal{D}_n; \theta, \mathbf{T})}.
\end{aligned}
\tag{29}
$$

The last equality holds since:

$$q(\mathbf{T}' \rightarrow \mathbf{T}) = q(\mathbf{T} \rightarrow \mathbf{T}'). \tag{30}$$

### C.2 ABOUT GROUP-FAIR CONSTRAINED POSTERIOR

In this subsection, we provide the table to show $\eta$ versus $\pi(\{\theta : \Delta_{\mathrm{DP}}(\theta) \leq \eta\}|\mathcal{D}_n)$, which was mentioned in Section 3.1. Here, $\Delta_{\mathrm{DP}}$ is defined with the threshold $\tau = 0$. Since the posterior distribution $\pi(\theta|\mathcal{D}_n)$ is intractable, we use Monte Carlo approximation to report those values. Specifically,

$$
\begin{aligned}
\pi(\{\theta : \Delta_{\mathrm{DP}}(\theta) \leq \eta\}|\mathcal{D}_n) &= \int_{\theta : \Delta_{\mathrm{DP}}(\theta) \leq \eta} \pi(\theta)\mathcal{L}(\mathcal{D}_n; \theta)\pi(d\theta) \\
&\approx \frac{1}{T}\sum_{t=1}^{T} \mathbb{I}(\Delta_{\mathrm{DP}}(\theta^{(t)}) \leq \eta),
\end{aligned}
\tag{31}
$$

where $(\theta^{(t)})_{t=1}^{T}$ are sampled from $\pi(\theta)\mathcal{L}(\mathcal{D}_n; \theta)$ using HMC with a learning rate 0.001. We used train dataset of ADULT as observations, and used a MLP with two hidden layers of width 200, as the model. We sampled 1,200 times, where first 200 samples are burn-in, and we thin the remaining samples at intervals of 10, resulting in 100 final samples. Here, original $\Delta_{\mathrm{DP}}$ in the train dataset is 0.1991, i.e. difference on the proportion of positive labeled individuals from each groups.

The results are given in Table 3 showing that: when $\eta$ drops below 0.10, the probability decreases to 0.03, making acceptance-rejection sampling practically infeasible.

Table 3: $\pi(\{\theta : \Delta_{\mathrm{DP}}(\theta) \leq \eta\}|\mathcal{D}_n)$ values for varying $\eta$.

| $\eta$ | 0.08 | 0.09 | 0.10 | 0.11 | 0.12 | 0.13 | 0.14 |
|---|---|---|---|---|---|---|---|
| Prob. | 0.00 | 0.02 | 0.03 | 0.05 | 0.07 | 0.08 | 0.10 |
| $\eta$ | 0.15 | 0.16 | 0.17 | 0.18 | 0.19 | 0.20 | 0.21 |
| Prob. | 0.12 | 0.14 | 0.16 | 0.20 | 0.53 | 0.85 | 1.00 |

### C.3 RANDOM-WALK METROPOLIS-HASTINGS

In Section 3.1, we explained why existing posterior sampling methods are challenging to implement with complex constraints such as group-fairness constraints. To provide empirical evidence for this statement, we conduct a simple experiment using the random-walk Metropolis-Hastings with constraints, which accepts only samples that satisfy a given group-fairness constraint, on ADULT. Using the train dataset, we propose a random walk proposals with Gaussian noise of learning rate 0.001. Then, we accepted proposals if random uniform variable $u \sim \mathrm{Uniform}(0, 1)$ is lower than the acceptance ratio $\alpha$ and the constraint is satisfied.

In Table 4, we can observe that as the constraint becomes stricter, the acceptance ratio decreases. Due to this low acceptance ratio, the efficiency of MCMC samples is significantly reduced. Fig. 5 is a corresponding illustration. Random proposals (yellow arrows) wander through regions of high posterior density, which could not satisfy the constraint level.

Table 4: Acceptance ratio of random-walk Metropolis-Hastings, only accepting samples that satisfies given constraint threshold.

| Constraint threshold | 0.18 | 0.175 | 0.17 | 0.165 | 0.16 | 0.155 |
|---|---|---|---|---|---|---|
| Acceptance ratio | 0.58 | 0.40 | 0.30 | 0.26 | 0.02 | 0.02 |

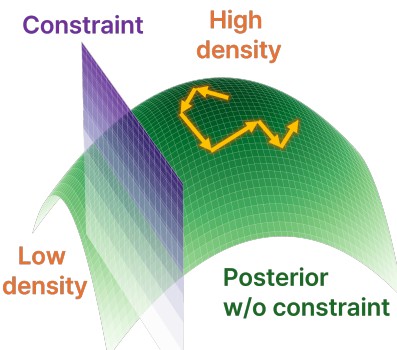

Figure 5: An illustration of a naive posterior sampling on a constrained parameter space. Yellow arrows indicate steps from a naive Metropolis-Hastings with random proposals, which wander through regions of high posterior density.

## C.4 OMITTED RESULTS

In this section, we provide all omitted Pareto-front lines of the main result from Section 5. From Figs. 6 and 7, we can observe that *gibbs_matched* shows better performance than other competitors in terms of performance-fairness trade-offs. For example, *gibbs_matched* achieves the best trade-off on all measures in ADULT with large margin. As mentioned in the main body, better Ece does not guarantee a good model. This is clearly observed with the result of adv, where its performance is clearly not favorable than others in terms of (Acc, Nll, brier), while maintaining good trade-off in Ece. To mitigate the increase of Ece under strict fairness levels, one can apply calibration techniques such as temperature scaling (Guo et al., 2017) without degrading Acc. We leave this to the future work.

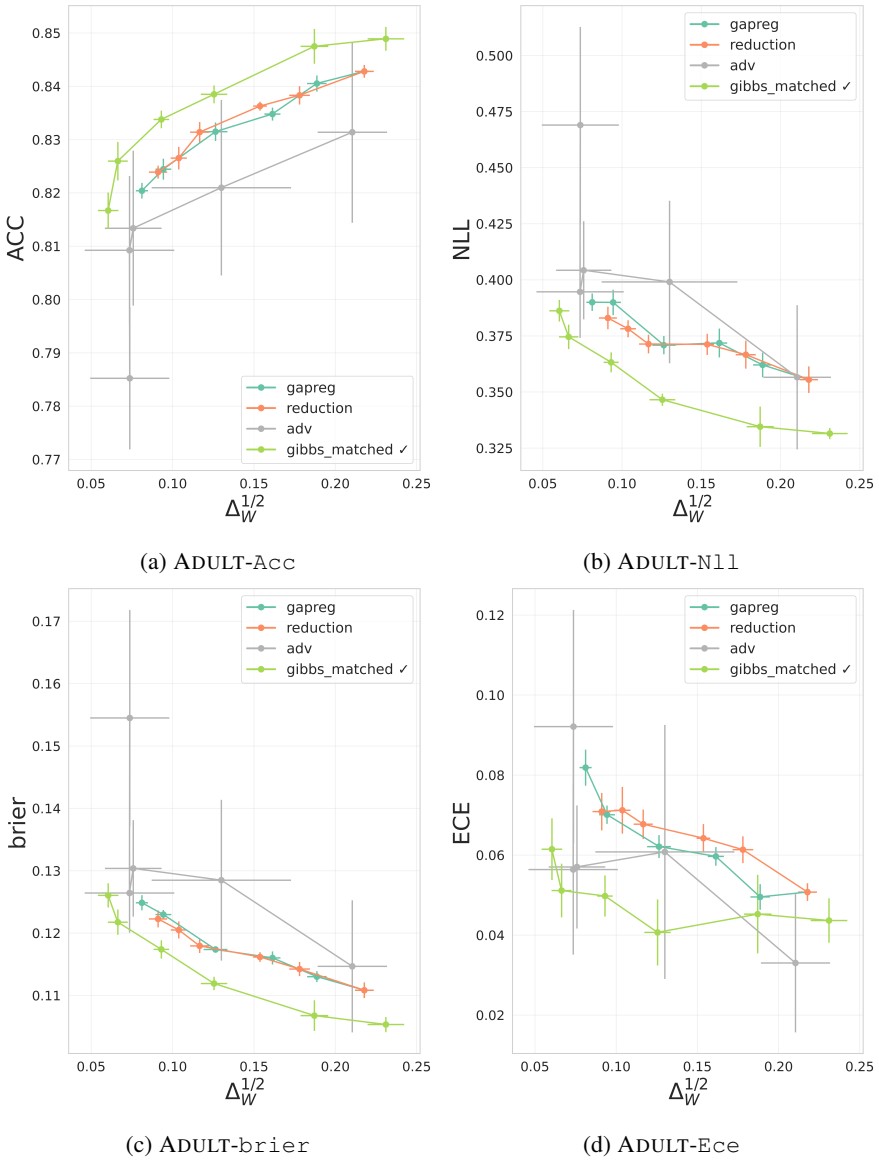

(a) ADULT-Acc

(b) ADULT-Nll

(c) ADULT-brier

(d) ADULT-Ece

Figure 6: ADULT. Pareto-front lines between level of $\Delta_W^{1/2}$ (on the $x$-axis) and the predictive performance (on the $y$-axis). The trade-off of proposed matched Gibbs posterior is superior to that of other competitors.

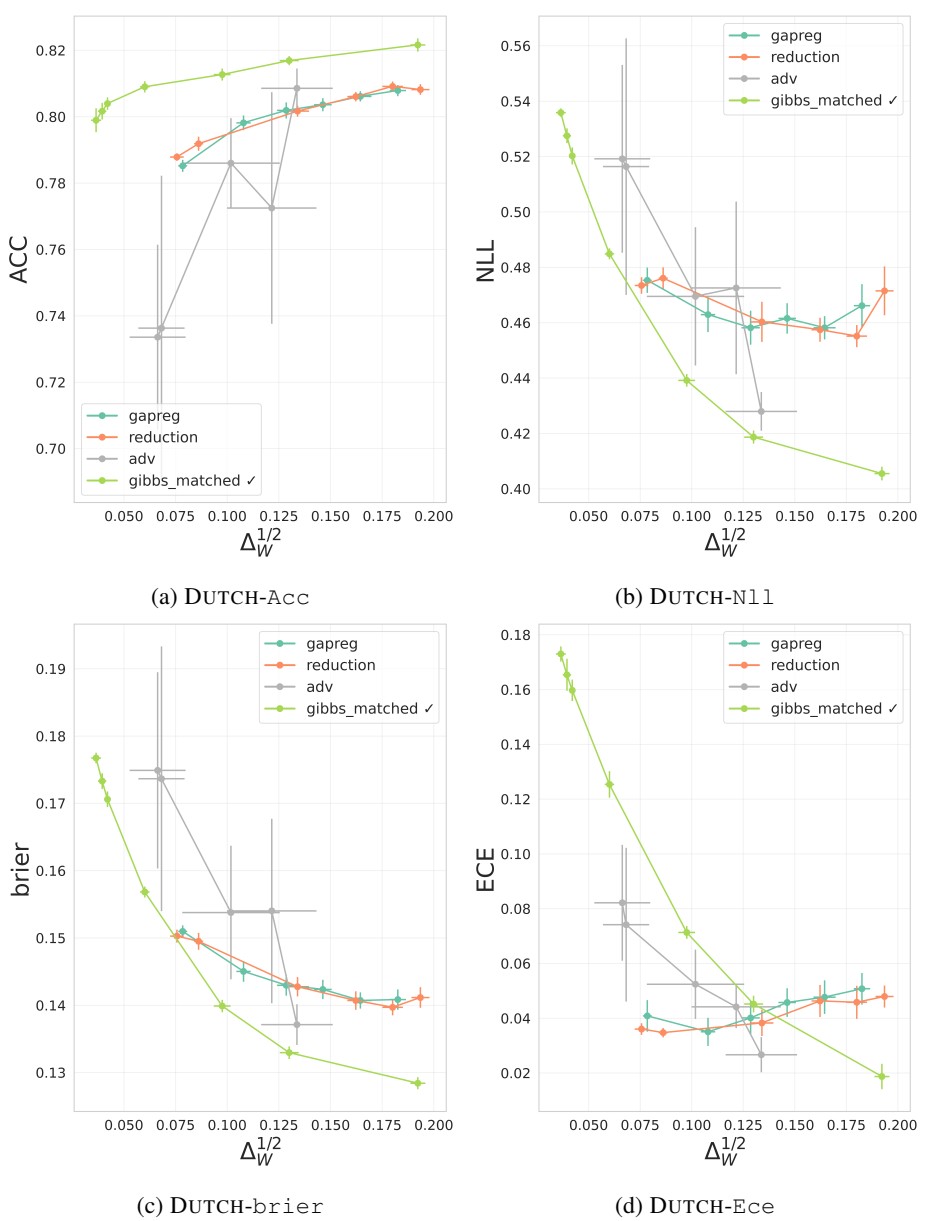

(a) DUTCH-Acc

(b) DUTCH-Nll

(c) DUTCH-brier

(d) DUTCH-Ece

Figure 7: DUTCH. Pareto-front lines between level of $\Delta_W^{1/2}$ (on the $x$-axis) and the predictive performance (on the $y$-axis). The trade-off of proposed matched Gibbs posterior is superior to that of other competitors.

## C.5 ABOUT STRONGLY $\psi$-FAIRNESS

**Strongly fair VI** Here, we empirically validate that the strongly fair variational inference with mean-field Gaussian does not work well, even for linear models. The experimental setup is similar to that of Appendix C.3. We simply optimize ELBO with a mean-field Gaussian distribution without any group-fairness aware constraints, on CRIME. For the model, we use a (linear) logistic regression model, with 100 epochs of training. We report the proportion of samples $f$ that are fair with level $\delta$, i.e. $\Delta_{\text{DP}}(f) \leq \delta$. Note that the original $\Delta_{\text{DP}}$ difference in the train dataset is $0.2642$. Table 5 shows that when $\delta$ becomes slightly smaller, the proportion of samples $f$ that satisfies the group fairness constraint vanishes. For the level of $0.24$ the proportion is only $1\%$ of the samples, and it totally vanishes to zero for the level under $0.22$. Hence, as mentioned in Section 3.2, variational inference with the mean-field Gaussian models cannot be directly applied to find a strongly fair distributions.

Table 5: Proportion of samples from mean-field Gaussian distribution optimized with standard ELBO, under varying level of $\delta$.

| $\delta$ | 0.27 | 0.26 | 0.25 | 0.24 | 0.23 | 0.22 |
|---|---|---|---|---|---|---|
| Prop. | 0.014 | 0.007 | 0.003 | 0.001 | 0.001 | 0.000 |

**Rejection sampling** To show that the aforementioned rejection sampling in Section 3.2 works well, we conduct an experiment on ADULT. Note that the rejection sampling of $\nu^{(s)}$ through $\nu^{(w)}$ can be performed simply by accepting the samples from $\nu^{(w)}$ that satisfies $\Delta_{\psi}(\cdot) \leq \delta$. We take 1000 samples from matched Gibbs posterior with the same settings of the main result on ADULT, and perform a rejection sampling.

Assume that we want samples that satisfies $\Delta_{\text{W}}^{1/2} < 0.10$, which is a quite strict condition as shown in Fig. 6. By sampling $1,000$ samples from matched Gibbs posterior with $\lambda = 4.0$, we can accept $897$ samples that satisfies the condition, yielding $0.897$ of acceptance ratio in the rejection sampling. Hence, we can effectively utilize matched Gibbs posterior to directly obtain strongly fair models.

## C.6 ADVANTAGE OF MATCHED GIBBS POSTERIOR: IMPROVED INDIVIDUAL FAIRNESS

As previously noted, the matched deviation resembles the concept of individual fairness, since minimizing it is equal to making outputs from two (matched) individuals. In this section, we empirically demonstrate how matched Gibbs posterior achieves better individual fairness metrics than other algorithms, when the $\Delta_{\text{W}}^{1/2}$ level is similar. We use the consistency score (`Con`) (Zemel et al., 2013; Yurochkin et al., 2020; Yurochkin & Sun, 2021), which measures the similarity on the labels for similar individuals. For the computation, we utilized `aif360` package[6].

From Tables 6 and 7, we can observe that the corresponding `Con` of matched Gibbs posterior achieves the best along with the best `Acc`, supporting the advantage of using matched Gibbs posterior. (In CRIME, we report `Con` of *adv* with its lowest $\Delta_{\text{W}}^{1/2}$, due to its instability.)

Table 6: CRIME. With a fixed level of $\Delta_{\text{W}}^{1/2} \approx 0.09$, the level of `Con` and corresponding `Acc` are reported. The **bold** faced values implies the best values.

| | | Con (Acc) | | | |
|---|---|---|---|---|---|
| *gapreg* | *reduction* | *adv* | *variational_mmd* | *gibbs_mmd* | *gibbs_matched* |
| 0.767 (0.718) | 0.768 (0.724) | 0.834 (0.684) | 0.790 (0.729) | 0.785 (0.696) | **0.837** (**0.736**) |

## C.7 MCMC DIAGNOSIS

We show that the MCMC from matched Gibbs posterior well-behaves in practice, which also considers **T** as a random variable. Following Betancourt (2017), we provide several plots of the en-

---

[6]`https://aif360.readthedocs.io/en/stable/modules/generated/aif360.sklearn.metrics.consistency_score.html`

Table 7: ADULT and DUTCH. With a fixed level of $\Delta_{\text{W}}^{1/2}$ (approximately 0.09 for ADULT and 0.07 for DUTCH, the level of Con and corresponding Acc are reported. The **bold** faced values implies the best Con.

| Dataset | Con (Acc) | | | |
|---|---|---|---|---|
| | *gapreg* | *reduction* | *adv* | *gibbs_matched* |
| ADULT | 0.927 (0.824) | 0.931 (0.824) | 0.954 (0.813) | **0.955** (**0.826**) |
| DUTCH | 0.939 (0.785) | 0.939 (0.788) | 0.960 (0.786) | **0.965** (**0.809**) |

ergy from each MCMC samples, that are commonly used to diagnose whether the MCMC is well-behaved. We also provide the energy Bayesian fraction of missing information (E-BFMI), a value that is larger than 0.3 indicates that the MCMC is not problematic (Betancourt, 2017). The definition of E-BFMI is as follows:

$$\widehat{\text{E-BFMI}} := \frac{\sum_{n=1}^{N}(E_n - E_{n-1})^2}{(E_n - \bar{E})^2} \tag{32}$$

Here, $E_n$ is an energy ($U + K$ in HMC) values of the $n$-th sample. For the diagnosis, we provide 3 energy related plots - traceplot of $E_n$, histogram of $\Delta E_n$, and lag plot $(E_{n-1}, E_n)$ - in Fig. 8, with results on ADULT. To validate the convergence, we provide the plots when the fairness level of the predictive distribution ($\mathbb{E}_\theta \Delta_{\text{W}}^{1/2}$) is strict ($\approx 0.09$). We can observe that the $E_n$ trace reduces first and then become stationary. Also, $\Delta E_n$ is well-distributed (not too sharp or wide, symmetry with center 0) and the lag plot is well-distributed as a point cloud of circle, indicating low autocorrelation, even when the group-fairness constraint is strict. In this case, $\widehat{\text{E-BFMI}} \approx 1.646 > 0.3$ holds, indicating that the MCMC well-behaves.

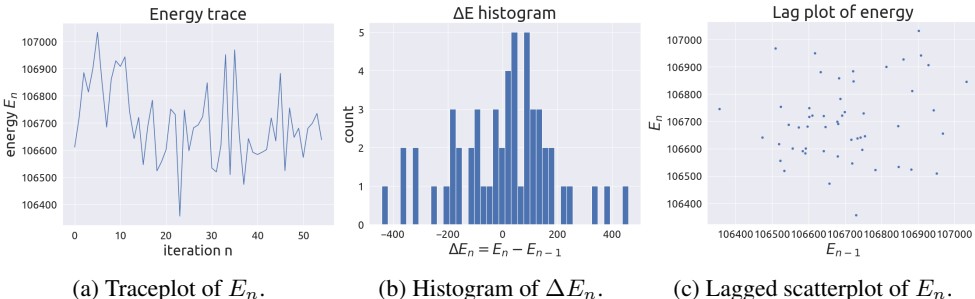

(a) Traceplot of $E_n$.  (b) Histogram of $\Delta E_n$.  (c) Lagged scatterplot of $E_n$.

Figure 8: Energy related plots of MCMC samples of matched Gibbs posterior on ADULT, when $\mathbb{E}_\theta \Delta_{\text{W}}^{1/2} \approx 0.09$.

We also measure the acceptance probability of $\mathbf{T}$ with different levels of $\lambda$. For $\lambda = (0.0, 0.5, 2.0, 3.0, 3.5, 4.0)$, corresponding acceptance ratio is $(0.38, 0.32, 0.3, 0.36, 0.38, 0.36)$, where the range of $[0.2, 0.5]$ is generally recommended in practice (Gelman et al., 1997; Roberts & Rosenthal, 2001). Hence, we conclude that our proposal of $\mathbf{T}$ for the Metropolis-Hastings is reasonable.

# D ABLATION STUDIES

## D.1 EFFECT OF $\tau$

In this section, we study the effect of $\tau$ to performances defined in the prior of $\mathbf{T}$, in Eq. (8). We report the performance of matched Gibbs posterior when $\tau$ is varying, on ADULT. Fig. 9 represents the trade-offs when the fairness level is strict, using $\tau \in \{0.01, 0.1, 1.0\}$. Table 8 is the detailed values of prediction performances. Note that smaller $\tau$ implies more powerful prior. We can observe that the values results in minimal variation. In other words, we can conclude that the choice of $\tau = 1.0$ does not significantly impact the main result. Hence, we can say that the belief from the observation is strong enough to overcome the effect of the prior, in the practical use of matched Gibbs posterior on ADULT.

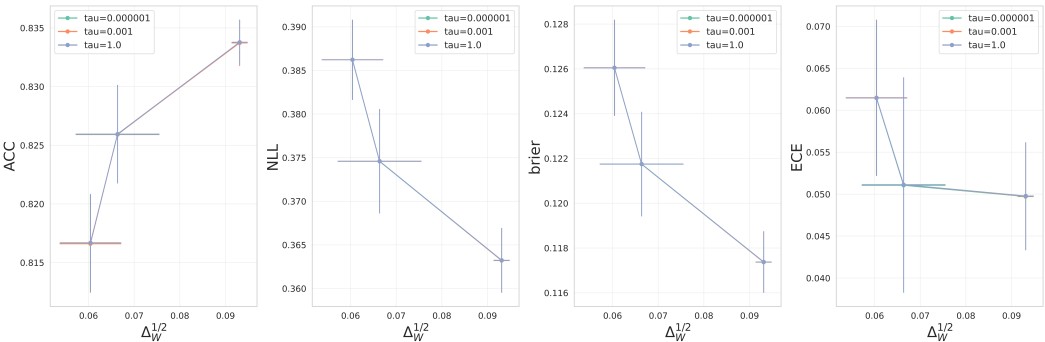

Figure 9: Pareto-front lines between level of group fairness $\Delta_{\mathrm{W}}^{1/2}$ and prediction performance of matched Gibbs posterior, with different values of $\tau$, on ADULT.

Table 8: Detailed values of the prediction performance with varying $\tau$, when $\Delta_{\mathrm{W}} \approx 0.09$ on ADULT.

| Measures | $\tau$ | | |
|---|---|---|---|
| | 0.01 | 0.1 | 1.0 |
| Acc $(\times 10^2)$ | 82.892 | 82.888 | 82.888 |
| Nll $(\times 10^2)$ | 36.462 | 36.463 | 36.463 |
| brier $(\times 10^2)$ | 11.860 | 11.860 | 11.860 |
| Ece $(\times 10^2)$ | 4.392 | 4.395 | 4.395 |

## D.2 EFFECT OF PRETRAIN EPOCHS

To induce the faster convergence, we utilized pretrained DNN parameters as initialization of the model parameters. We conduct an additional experiment by varying pretrain epochs on ADULT, to observe the effect on the performances. With the same setting from the main result, we report the Pareto-front lines with varying pretrain epochs as $[10, 20, 30, 40]$, and use the same values for remaining hyperparameters such as burn-in epochs. Note that we utilized 30-epoch case for the initialization in the main result.

Fig. 10 is the corresponding Pareto-front lines. Although quality of MCMC samples largely depends on the chain configuration, we can observe that the number of pretrain epochs does not significantly affect the predictive performance of matched Gibbs posterior when the chain is sufficiently iterated, which is not for the case for 10 epochs. The 10-epoch case shows large performance gaps, primarily because the MCMC was not run long enough. We can also observe that more pretrain epochs results better performance, primarily because the MCMC converges more quickly.

## D.3 EFFECT OF FLIPPED SENSITIVE ATTRIBUTES

Matched Gibbs posterior utilizes the matched deviation defined in Eq. (6). When the number of individuals from each sensitive group are different, the matched deviation changes when the labels

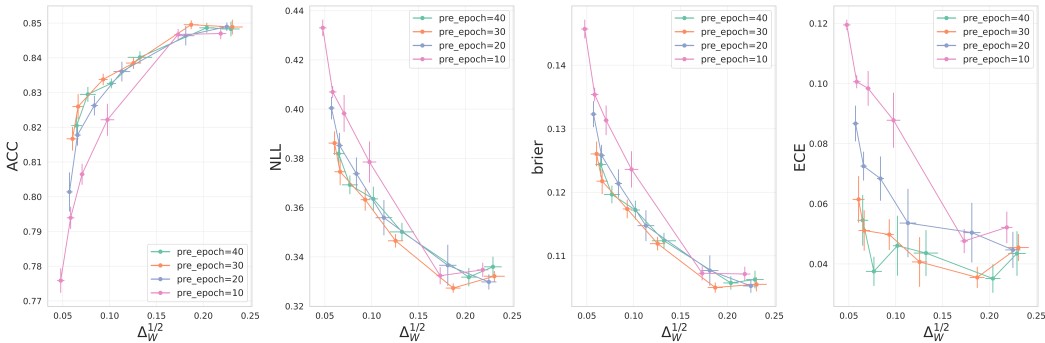

Figure 10: Pareto-front lines between level of group fairness $\Delta_W^{1/2}$ and prediction performance of matched Gibbs posterior, with varying pretrain epochs, on ADULT.

of sensitive attributes are flipped. Practically, we construct the matching function $\mathbf{T}$ from the smaller sensitive group to the larger sensitive group. To observe the effect of the flipping, we conduct an experiment on ADULT with the same settings from the main result.

In Fig. 11, we can observe that flipping the sensitive attribute (orange line) does not lead to a significant change in performance. The slight decrease in performance can likely be explained by the fact that matching from the larger group naturally permits many-to-one matching, which may overly reduce the distance between individuals from $S = 0$ and $S = 1$. In contrast, matching from the smaller group does not exhibit this issue.

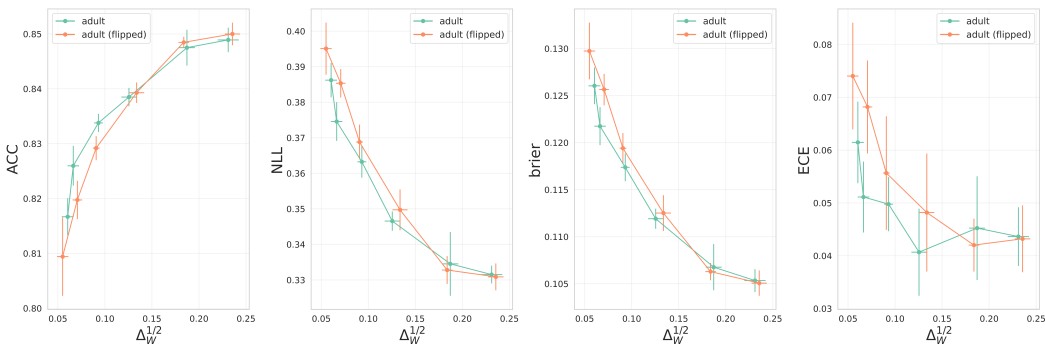

Figure 11: Pareto-front lines between level of group fairness $\Delta_W^{1/2}$ and prediction performance of matched Gibbs posterior, with flipped sensitive attributes, on ADULT.

## D.4 IMPACT OF PRIOR CHOICE

The proposed matched Gibbs posterior does allow the different priors to be considered. We investigate how variations in the prior distribution influence the model's inference. While the main analysis focuses on the Gaussian prior, we additionally examine alternative priors to show the robustness of our method. We conduct experiments using the CRIME dataset and employ Gaussian, Cauchy, and Student-t priors to evaluate how different prior specifications affect the model's performance. Note that Cauchy and Student-t priors are alternatively considered when heavy-tailed priors are needed (Fortuin, 2022).

In Fig. 12, we can observe that the performances does not significantly change when different priors are considered.

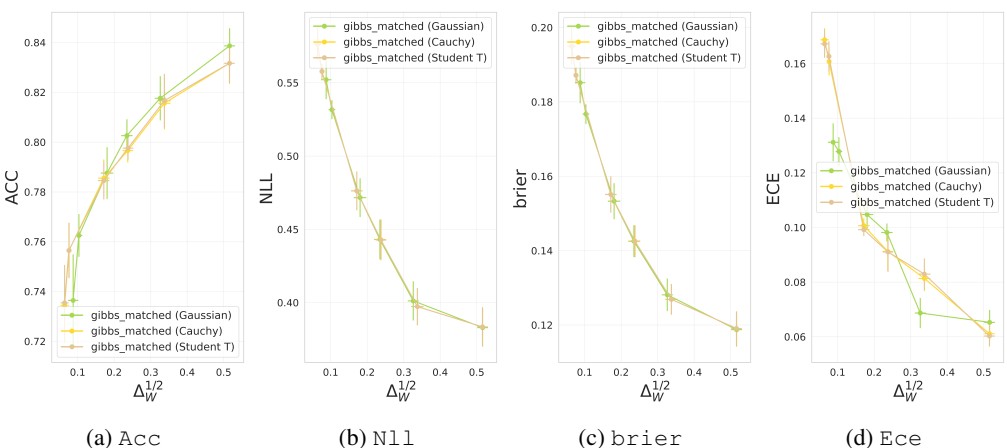

(a) Acc  (b) Nll  (c) brier  (d) Ece

Figure 12: Pareto-front lines between level of group fairness $\Delta_W^{1/2}$ and prediction performance of matched Gibbs posterior with varying priors, on CRIME.

# E    ADDITIONAL RESULTS

## E.1    EXTENSION TO EQUALIZED ODDS

We mainly focus on demographic parity as a fairness measure. In this section we show that the extension with regard to EO (equalized odds) can be done similar.

Specifically, we consider two matching functions $\mathbf{T}_0 : \mathcal{X}_{0,1} \to \mathcal{X}_{0,0}$ and $\mathbf{T}_1 : \mathcal{X}_{1,1} \to \mathcal{X}_{1,0}$, where $\mathcal{X}_{y,s}$ is the domain of $X|Y = y, S = s$. Now, we can modify the matched deviation as

$$
\begin{aligned}
\Delta_{\mathrm{M}}(\theta, \mathbf{T}_0, \mathbf{T}_1) :=& p_{0|1} \mathbb{E}_{X_1 \sim \mathbb{P}_{(0,1)}}(\|f_\theta(X_1, s = 1) - f_\theta(\mathbf{T}_0(X_1), s = 0)\|^2) \\
& + p_{1|1} \mathbb{E}_{X_1 \sim \mathbb{P}_{(1,1)}}(\|f_\theta(X_1, s = 1) - f_\theta(\mathbf{T}_1(X_1), s = 0)\|^2)
\end{aligned}
\tag{33}
$$

where $p_{y|s} = \Pr(Y = y|S = s)$ and $\mathbb{P}_{(y,s)}$ is a conditional distribution of $X|Y = y, S = s$. Now we can define matched Gibbs posterior with respect to the modified matched deviation.

For the MCMC, we consider MH for $(\mathbf{T}_0, \mathbf{T}_1) \sim p(\mathbf{T}_0, \mathbf{T}_1|\theta, \mathcal{D}_n)$. We can consider a prior $e(\mathbf{T}_0, \mathbf{T}_1) := \exp\left(-\frac{1}{n_{01}} \sum_{i=1}^{n_{01}} d(X_i^{(0,0)}, \mathbf{T}_0(X_i^{(0,1)})) - \frac{1}{n_{11}} \sum_{i=1}^{n_{11}} d(X_i^{(1,0)}, \mathbf{T}_1(X_i^{(1,1)}))/\tau\right)$. For the proposal, we can reuse the partially permuted proposal of Eq. (9) for $y \in \{0, 1\}$ as:

$$
\mathbf{T}'_y(j) := \begin{cases} \mathbf{T}_y(j) & \text{for } j \notin [n_{y1}] \setminus \{i_1^y, \ldots, i_k^y\} \\ \mathbf{T}_y(i_{\Pi_k(l)}^y) & \text{for } j = i_l^y \end{cases}
\tag{34}
$$

Using this formulation, we additionally perform an experiment on CRIME, to further show that the trade-offs from matched Gibbs posterior with regard to EO are better than the baseline methods, in Fig. 13. The results are similar to the case of DP, strengthening the ability of matched Gibbs posterior.

We can continuously observe that matched Gibbs posterior shows better performances than other baselines. Therefore, we believe that the proposed matched Gibbs posterior can also be well applied to EO.

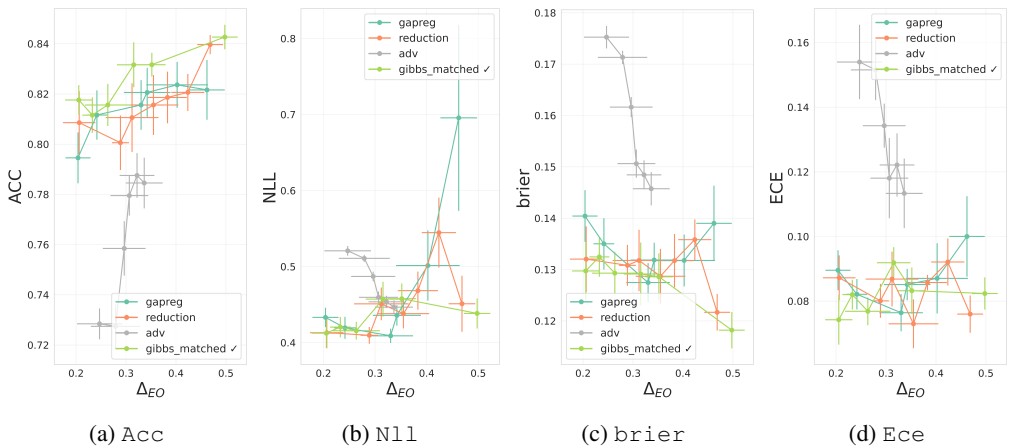

|(a) Acc|(b) Nll|(c) brier|(d) Ece|

Figure 13: Pareto-front lines between level of group fairness $\Delta_{\mathrm{EO}}$ and prediction performance of matched Gibbs posterior, on CRIME.

## E.2    EXTENSION TO MULTINARY SENSITIVE ATTRIBUTE CASE

We can easily extend matched Gibbs posterior into the case of multinary sensitive attribute, $S \in \{0, 1, \ldots, M\}$ for an integer $M > 1$. Specifically, we may choose one sensitive group (for example, $\mathcal{X}_0$) as an anchor group and define matching functions $\mathbf{T}_i : \mathcal{X}_i \to \mathcal{X}_0$ that map each of the other sensitive groups to this anchor group. In this setting, we can define the matched deviation as $\Delta_{\mathrm{M}}(\theta, (\mathbf{T}_i)_{i=1}^M) := \sum_{i=1}^M \mathbb{E}_{X_i \sim \mathbb{P}_i}(\|f_\theta(X_i, s = 1) - f_\theta(\mathbf{T}_i(X_i), s = 0)\|^2)$, and then construct

matched Gibbs posterior using this quantity. For MCMC procedure, the extension is also straight-forward: we can simply apply the Gibbs sampler iteratively for each $\mathbf{T}_i$, using same MH proposal with Eq. (9).

### E.3 PERFORMANCE BY STRONG DEMOGRAPHIC PARITY

For an additional support of the performance of matched Gibbs posterior, we provide the trade-offs measured with other DP metric, Strong Demographic Parity (SDP). The definition of SDP gap is as follows (Jiang et al., 2020a; Silvia et al., 2020):

$$\Delta_{\mathrm{SDP}}(f) := \mathbb{E}_\tau \left| \Pr(f(X, S = 0) \geq \tau) - \Pr(f(X, S = 1) \geq \tau) \right|. \tag{35}$$

This is called **strong** DP, since this is equivalent with $\mathbb{E}_\tau \Delta_{\mathrm{DP}}(f)$, which enforces DP to be hold regardless of the threshold value $\tau$. In other word, enforcing SDP does not depend on the threshold value, and thus not sensitive to the choice of the threshold (Jiang et al., 2020a; Silvia et al., 2020).

We evaluate the utility-fairness / uncertainty-fairness trade-offs, using the original DP as a fairness measure, on CRIME dataset on the same setting of the main experiment. The results are in Fig. 14. We can consistently observe that the trade-offs of matched Gibbs posterior are better than the baseline methods.

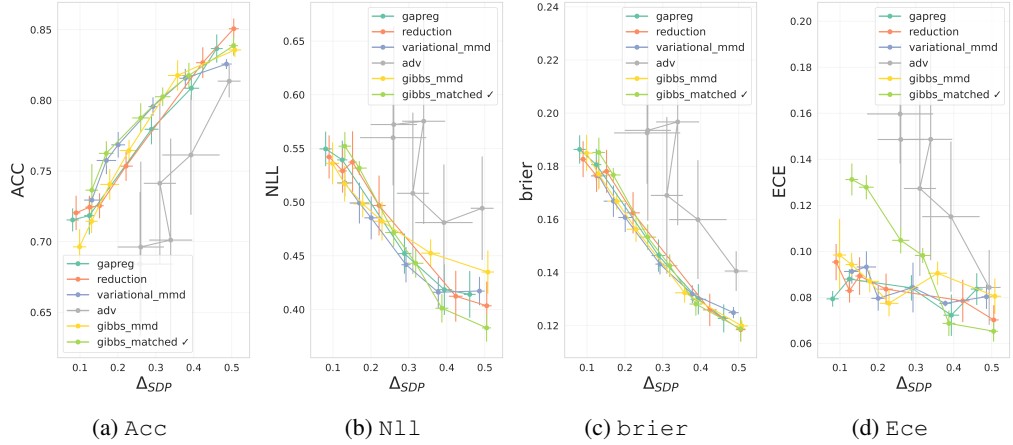

(a) Acc   (b) Nll   (c) brier   (d) Ece

Figure 14: Pareto-front lines between level of group fairness $\Delta_{\mathrm{SDP}}$ and prediction performance of matched Gibbs posterior, on CRIME.

### E.4 ABOUT COMPUTATIONAL COSTS

The computational complexity of the MH step for $\mathbf{T}$ does not depend on the input dimension, and its scale on each step is $O(1)$ since it is a random permutation on a fixed window size of $k$. We provide a running time table of the main experiment on CIVIL dataset, in Table 9. We can observe that the time spent on the actual sampling (for joint inference of $f$ and $\mathbf{T}$) is relatively small.

This is because (i) the scale on each step is small as mentioned, and (ii) since we designed the procedure to use a pretrained model as a good initial point for $f$ and OT-based initialization for $\mathbf{T}$, the MCMC converges quickly and reliably. See Appendix C.7 for additional supporting evidences of the reliability of the proposed MCMC.

### E.5 ADDITIONAL DATASET

There are some concerns regarding the use of ADULT dataset (Ding et al., 2021a). Therefore, we additionally employed the ACSINCOME dataset which is an alternative to ADULT, to reproduce the experiments presented in the main text. Following the experimental setup in (Han et al., 2023), we employed a MLP with four hidden layers of width 128, and used *gapreg* and *reduction* for 10 epochs. To give an efficient initial value, we utilized pretrained DNN models as initial parameter values, with pretrain epoch 10. The remaining experimental settings are provided in Appendix B.2.

Table 9: A running time table on CIVIL dataset. Etc. denotes the time measurement on initializations.

| Method | Time (sec.) | Etc. (sec.) |
|---|---|---|
| *gapreg* | 48.85 | |
| *reduction* | 47.32 | |
| *gibbs_matched* | 24.49 | pretrain: 46.87 ot initial: 48.56 |

We split the dataset into two partitions with a proportion $8 : 2$, corresponding to train and test datasets. Table 10 provides the basic descriptions of the ACSINCOME dataset.

Table 10: Summary statistics for ACSINCOME dataset.

| Dataset | Features ($d$) | Size | $Y = 1$ | $S = 1$ | train samples | test samples |
|---|---|---|---|---|---|---|
| ACSINCOME | 120 | 1,664,500 | 614,067 | 866,735 | 1,331,600 | 332,900 |

The corresponding results in Fig. 15. Since the adversarial debiasing method (Adv) (Zhang et al., 2018) showed worse performance compared to the baseline and our method, we excluded it from this experiment. We can consistently observe that the trade-offs of *gibbs_matched* is better than the other algorithms.

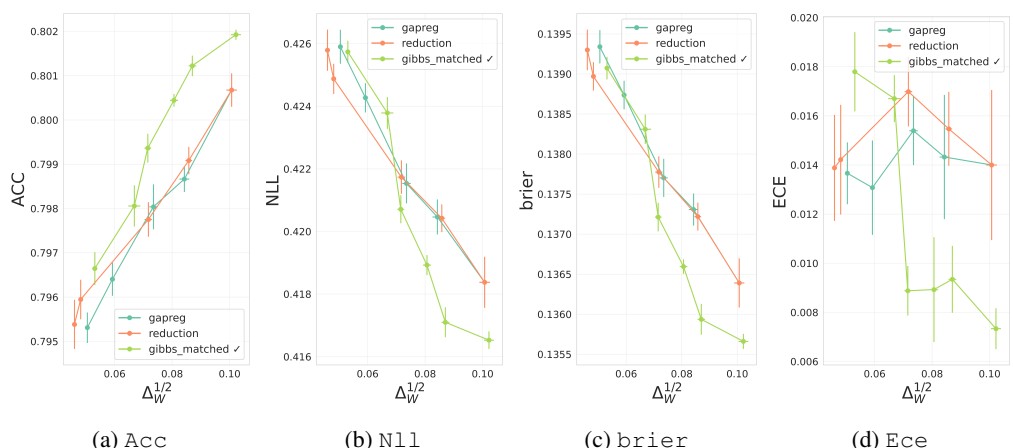

(a) Acc      (b) Nll      (c) brier      (d) Ece

Figure 15: Pareto-front lines between level of group fairness $\Delta_W^{1/2}$ and prediction performance of matched Gibbs posterior, on ACSINCOME.

