# OpenReview forum: "A Fair Bayesian Inference through Matched Gibbs Posterior"
_ICLR.cc/2026/Conference — ICLR 2026 Poster_

### Official Review · Reviewer_ZbGT · 2025-10-28

**Soundness:** 3
**Presentation:** 4
**Contribution:** 2
**Rating:** 6
**Confidence:** 3

**Summary:**

The paper introduces a Bayesian framework for fair machine learning, addressing group fairness and uncertainty quantification  simultaneously. Traditional group-fair models ensure fairness by constraining optimization objectives but rarely account for predictive uncertainty, which is crucial for trustworthy AI. The authors propose a matched Gibbs posterior, derived from a novel fairness-aware penalty called the matched deviation, which upper-bounds the Wasserstein and total-variation fairness measures. This formulation avoids adversarial training and provides a computationally efficient approximation to the fair Bayesian posterior. They design an MCMC algorithm where both model parameters and the matching function 𝑇 are inferred jointly, ensuring fairness without heavy computation. Experiments on tabular, image, and text datasets show that the matched Gibbs posterior achieves better trade-offs between utility vs fairness and uncertainty vs fairness than baselines such as Reduction, GapReg, and Adversarial fairness methods, while also improving individual fairness.

**Strengths:**

1. Well written and clearly structured: Despite heavy mathematics, the paper is well organized; intuitions precede theorems, notation is consistent, and experiments visually support claims.

2. Novel combination of fairness and Bayesian inference: The paper is among the first to integrate group fairness constraints directly into Bayesian inference, explicitly addressing both uncertainty quantification and fairness.

3. Avoids adversarial optimization: By replacing adversarial discriminators with a learnable matching function 𝑇, the approach sidesteps instability and 𝑂(𝑛^2) cost typical in IPM-based fairness.

4. Improved fairness–utility trade-off: Across datasets, matched Gibbs posterior consistently outperforms prior baselines on accuracy, NLL, Brier, and calibration error (ECE).

**Weaknesses:**

1. Scalability concerns: Joint inference of 𝑓 and 𝑇 may become costly for high-dimensional or non-metric input spaces (e.g., text embeddings). No discussion on large-scale efficiency.

2. Restricted to binary sensitive attributes: The method currently handles only 𝑆∈{0,1}; multi-group or intersectional fairness remains unexplored.

3. Empirical scope and baselines: Experiments are thorough but limited to medium-sized datasets; modern large-scale deep architectures (e.g., BERT, ResNet-50) are absent.

4. Unclear robustness under complex priors: The framework assumes tractable Gaussian priors; how matched Gibbs behaves with non-Gaussian or hierarchical priors is not tested.

My main concerns focus on scalability and applicability to modern large-scale architectures. Specifically, the paper lacks validation on high-capacity models such as BERT or ResNet-50, and it remains unclear whether the proposed matched Gibbs posterior remains computationally feasible in high-dimensional or non-metric spaces (e.g., text embeddings). In such settings, stability and sensitivity to noise could become significant issues.

**Questions:**

Impact of imperfect matching: How sensitive is fairness performance to suboptimal or noisy matching functions 𝑇? Can the authors quantify how deviation from optimal 𝑇 affects fairness bounds?

Scalability: How would the proposed MCMC perform on large neural models (e.g., transformers) or high-dimensional text embeddings?

Connection to uncertainty geometry: Can the authors relate matched Gibbs fairness constraints to curvature of the posterior (e.g., Hessian eigen structure) or persistent-homology-based fairness landscapes? (This is not necessary for the paper, just something that came to mind during the review.)

---

> ### Author Response · Authors · 2025-11-20
>
> Thank you for the detailed feedback and thoughtful inquiries. We have carefully responded to each of the issues you noted. In the revised paper, we marked the differences and added results in blue. Please note that the figure and section numbers mentioned in the comments correspond to those in the **revised version** of the paper.
>
> # Summary of changes
> Here is a summary of the changes made in the revised version.
> * Section 1: We revised the introduction to emphasize our motivation and contributions.
> * Appendix E.1: We explain the extension of matched Gibbs posterior with respect to Equalized Odds, along with additional experimental results.
> * Appendix E.2: We explain the extension of matched Gibbs posterior to the case of multinary sensitive cases.
> * Appendix E.3: We report additional experimental results, where the fairness metric is evaluated with different metric, strong demographic parity (SDP).
> * Appendix E.4: We report additional experimental results, when the prior choice varies.
> * Appendix E.5: We further discuss about the computational cost of matched Gibbs posterior.
> * Appendix E.6: We report additional experimental results, on an additional dataset, **ACSIncome**.
>
> # Responses
> > W1. Scalability concerns...
> * The method does not significantly affected by the dimension of inputs. The only computational difference on matched Gibbs posterior and the original posterior is the calculation of matched deviation $\Delta_\mathrm{M}$. However, the calculation of matched deviation is equal to a simple Euclidean distance calculation, which does not heavily affected by the dimension of inputs.
>
> * **(Empirical support)** We conducted experiments on the **CelebA** image dataset and the **Civil** text dataset to demonstrate large-scale efficiency, and in both cases, matched Gibbs posterior exhibited a relatively better trade-off. In addition, we note that the method includes a hyperparameter $k$, which controls the window space of the $\mathbf{T}$ proposal, allowing the procedure to be tuned appropriately for each dataset.
>
> > W2. Restricted to binary sensitive attributes...
> * For the multinary sensitive attribute case ($S\in\{0,1,\ldots,M\}$ for an integer $M>1$), as mentioned in our paper, the method can be extended by borrowing the approach from paper [1]. More specifically, we may choose one sensitive group (for example, $\mathcal{X}_0$) as an anchor group and define matching functions  $\mathbf{T}_i:\mathcal{X}_i\to\mathcal{X}_0$ that map each of the other sensitive groups to this anchor group.
>
> * In this setting, we can redefine the matched deviation as the sum of matched deviation with respect to $\mathbf{T}_i$, and then construct matched Gibbs posterior using this quantity.
>         For MCMC procedure, the extension is also straightforward: we can simply apply the Gibbs sampler iteratively for each $\mathbf{T}_i$. Please refer to **Appendix E.2** for more detailed discussion on this extension.
>
> * Furthermore, when multiple sensitive attributes are considered simultaneously, resulting in a number of intersectional subgroups, one can also treat the problem in a multinary manner.
>
> * **(A future direction)** On the other hand, if the number of sensitive attributes is very large, this may lead to high computational complexity and/or data sparsity issues. We view this as an interesting direction for future research, and we plan to pursue it in the near future. Thank you again for this helpful suggestion.

---

> ### Author Response · Authors · 2025-11-20
>
> > W3. Empirical scope and baselines...
> * First, training such a large model in a fully Bayesian manner is itself a challenging problem that needs to be addressed. There are several papers that aim to solve this issue [2,3,4]. For the inference of matched Gibbs posterior, we can utilize these methods.
> * Second, as in previous researches, it is standard to use pretrained representations and then attach a head network for the downstream task (e.g. [5]). For example, the **Civil** dataset we used is a benchmark for toxicity classification, and prior studies also rely on embeddings. Likewise, as described in Appendix B.1, we used pretrained RoBERTa embeddings for **Civil**.
> * Using the representation of a backbone model to perform a downstream task is a common trend in recent work (e.g. CLIP or LLaVa). Our method follows this same approach, so it would be helpful to understand our focus in that context.
> * Developing a new and efficient approach to address this issue appears to be a promising research direction. One possible approach would be to combine Bayesian Low-rank adaptation [6] with this methodology. Since this method introduces additional learnable parameters for large-scale deep architectures and performs Bayesian inference over them, it seems likely that it can be effectively integrated with matched Gibbs posterior.
> * We are conducting an experiment related to this approach, to determine whether sampling additional parameters from matched Gibbs posterior can be done successfully with varying $\lambda$ values. However, due to the scale of the model and various technical difficulties, it is not certain whether we will be able to report results within the discussion period. We appreciate your insightful comments, which have helped us identify directions for further development.
>
> > W4. Unclear robustness under complex priors
> * Note that the Gaussian prior is not conjugate with deep neural networks, and so
>         we use MH algorithms. Thus, any priors can be used without much modification.
>         To validate this, we conducted additional experiments using Cauchy and Student-t priors. The results are provided in **Figure 14 of Appendix E.4.**
>         We can observe that the performances does not significantly change when different priors are considered.
>
> * It is quite standard to use a Gaussian prior, as it is the most practically convenient choice, especially for variational inference. In this regard, our method offers an additional advantage. By reformulating variational inference as the problem of drawing samples from the matched Gibbs posterior, our approach can be implemented using MCMC even when non-Gaussian priors are employed. However, if one wishes to work within the standard variational inference framework, using a non-Gaussian prior would require additional discussion on how to compute the ELBO, which serves as the optimization objective.
>
> * In addition, we have already conducted an ablation study on the prior hyperparameter ($\tau$) in **Appendix D.1.** The results show that as the influence of the likelihood becomes sufficiently large and the effect of the prior becomes negligible. This is consistent with the well-known principle in Bayesian inference that, as the number of observations increases, the likelihood begins to dominate the posterior.
>
> > W5. My main concerns focus on scalability and applicability to modern large-scale architectures...
> * Please see our response to Weakness 3.
>
> ## References
> [1] Fair clustering via alignment. Kim et. al. ICML. 2025.
>
> [2] Bayesian learning via stochastic gradient langevin dynamics. Welling and Teh. ICML. 2011.
>
> [3] Stochastic gradient hamiltonian monte carlo. Chen et. al. ICML. 2014.
>
> [4] Bayesian sampling using stochastic gradient thermostats. Ding et. al. NeurIPS. 2014.
>
> [5] On learning fairness and accuracy on multiple subgroups. Shui et. al. NeurIPS. 2022.
>
> [6] Bayesian Low-rank Adaptation for Large Language Models. Yang et. al. ICLR. 2024.

---

> > ### Author Response · Authors · 2025-11-21
> >
> > > Q1. Impact of imperfect matching...
> > * An interesting property is that the fairness is guaranteed for any one-to-one matching function as we have proved in Theorem 4.1. When the sample sizes of two sensitive groups are the same, and we only consider an one-to-one matching function, the fairness level is guaranteed and thus our mission is to find a good matching function for prediction accuracy and the proposed MCMC algorithm does do it for exploring the optimal $f$ and $T,$ simultaneously.
> >
> > * When the sample sizes are not equal, there is no one-to-one matching function. We practically construct $T$ from the smaller sensitive group to the larger sensitive group. We provide an empirical result on the flipped version of $T$ in **Appendix D.3.** We can observe that the fairness level changes only minimally; rather, it influences the utility or uncertainty achievable under that fairness level.
> >
> > > Q2. Scalability: How would the proposed MCMC perform on large neural models (e.g., transformers) or high-dimensional text embeddings?
> >
> > * Please see our response to Weakness 1 and 3.
> >
> > > Q3. Connection to uncertainty geometry...
> >
> > * We think that the constraint incorporated in the matched Gibbs posterior is unlikely to substantially affect the curvature of the posterior. For example, in the Gaussian regression problem discussed in **Appendix A.4**, the matched Gibbs posterior remains a Gaussian process. This suggests that the posterior curvature may not change significantly under the constraint.
> >
> > * Additionally, assume that we use a deviation such as MMD to construct posterior instead of $\Delta_\mathrm{M}$. Then the exponential term appearing in the definition of MMD would in turn be nested inside another exponential, which would likely have a substantial impact on the curvature. In contrast, by working with the matched deviation, we can see in cases like the one in Appendix A.4 that the curvature is designed not to be heavily affected. This may (conjecturally) be one reason why the Gibbs posterior defined via the matched deviation achieves such empirical success.

---

> > > ### Comment · Reviewer_ZbGT · 2025-11-26
> > >
> > > Thank you for working to address the questions raised. I maintain my current score.

---

### Official Review · Reviewer_vJDv · 2025-11-01

**Soundness:** 3
**Presentation:** 3
**Contribution:** 2
**Rating:** 4
**Confidence:** 4

**Summary:**

The paper develops a Bayesian route to group fairness by defining fairness for posteriors and proposing a matched Gibbs posterior: a Gibbs posterior with a new matched deviation penalty that avoids adversarial IPM/MMD inner loops, comes with bounds relating it to Wasserstein/TV, and is trained via an HMC+MH sampler that jointly infers the predictor and a matching function $T$. Experiments on ADULT, DUTCH, CRIME, CELEBA, and CIVIL suggest improved utility–fairness and uncertainty–fairness trade-offs.

**Strengths:**

1. Clear formulation of fairness for posteriors (average DP to strong DP via rejection) and a practical proxy—the matched Gibbs posterior—that circumvents adversarial critics and offers $O(n)$ updates with an explicit sampler.
2. Sound theory + strong experiments. Bounds linking matched deviation to Wasserstein/TV give intuition, and the image/text/tabular results show stronger Pareto fronts.

**Weaknesses:**

1. Fairness target & positioning. The work focuses on demographic parity. Is there any justification for this choice against other metrics (Equal opportunity/equalized odds/calibration)?
2. Fairness is measured on score distributions (W2(P_{f,0}, P_{f,1})); the relation to thresholded decisions (rate gaps) is unclear. Could you provide a more detailed analysis connecting score-level DP to rate-level DP, with sensitivity to thresholds or post-hoc calibration?
3. The prior and MH proposal for T (swap k matches) are reasonable, but (i) what distance d is used for images/text (esp. CIVIL)? (ii) How does mixing/acceptance scale with n, class imbalance, and high-dimensional X?
4. Complexity claims. The paper argues $O(n)$ per update vs. $O(n^2)$ MMD; it would be better if there were some running time/memory tables on large splits and report the cost of the HMC step (leap-frogs, step size) and MH over $T$.
5. Concerns about the dataset. The empirical studies are very strong. However, the adult dataset has some known issues (https://arxiv.org/abs/2108.04884), it would be better to try out a more robust dataset or mention these caveats.
6. Limitations. The current proposed method seems only applicable to binary sensitive attributes. Are there any discussions on how to extend to multi-categorical/continuous ones? Also, how could it be generalized to conditional fairness notions like equalized odds?

I'm willing to raise the score if my concerns are discussed and addressed, thank you.

**Questions:**

Please see weakness.

---

> ### Author Response · Authors · 2025-11-20
>
> We appreciate your thorough reviews and insightful questions, and we have worked hard to address all of your comments and concerns. In the revised paper, we marked the differences and added results in blue. Please note that the figure and section numbers mentioned in the comments correspond to those in the **revised version** of the paper.
>
> # Summary of changes
> Here is a summary of the changes made in the revised version.
> * Section 1: We revised the introduction to emphasize our motivation and contributions.
> * Appendix E.1: We explain the extension of matched Gibbs posterior with respect to Equalized Odds, along with additional experimental results.
> * Appendix E.2: We explain the extension of matched Gibbs posterior to the case of multinary sensitive cases.
> * Appendix E.3: We report additional experimental results, where the fairness metric is evaluated with different metric, strong demographic parity (SDP).
> * Appendix E.4: We report additional experimental results, when the prior choice varies.
> * Appendix E.5: We further discuss about the computational cost of matched Gibbs posterior.
> * Appendix E.6: We report additional experimental results, on an additional dataset, **ACSIncome**.
>
> # Responses
> > W1. Fairness target \& positioning. The work focuses on demographic parity...
> * We focus on DP for the sake of simplicity. However, we can easily extend our proposed matched Gibbs posterior to handle equalized odds. Please refer to **Appendix E.1** for the detailed formulation and corresponding MCMC procedure.
> Using this formulation, we additionally perform an experiment to yield utility-fairness and uncertainty-fairness trade-offs.
> We can continuously observe that matched Gibbs posterior shows better performances than other baselines. Therefore, we believe that the proposed matched Gibbs posterior can also be well applied to equalized odds.
>
> > W2. Fairness is measured on score distributions...
> * Discussions on this topic are already addressed in prior works [1,2]. Enforcing rate-level DP is highly sensitive to the choice of threshold and does not guarantee that DP will hold for other thresholds. Moreover, using an indicator function for class prediction introduces optimization difficulties. For these reasons, works such as [1,2] have adopted score-level DP, and we also follow this direction by considering score-level DP, which is more suitable for the proposed fair variational Bayesian inference framework.
>
> * Consider Strong Demographic Parity (SDP) [1,2], as a representative example of score-level DP:
>         $\Delta_\mathrm{SDP}(f)=\mathbb{E}_{\tau}\left\lvert \Pr(f(X,S=0)\ge\tau)-\Pr(f(X,S=1)\ge\tau) \right\rvert.$
> * Here, $\tau$ is a random variable following the uniform distribution ot Gaussian distribution depending on the definition of $f.$ We can observe that $\Delta_\mathrm{SDP}$ (score-level DP) is a stronger version of $\Delta_\mathrm{DP}$ (rate-level DP), which enforces the decision does not depend on the sensitive regardless of the threshold value $\tau$.
>
> * To support the empirical success of matched Gibbs posterior, we additionally measured the trade-offs from the main text using the above SDP gap. The results are provided in **Figure 13 in Appendix E.3.** We can consistently observe that the performance of matched Gibbs posterior is better than the competitors.
>
> > W3. The prior and MH proposal for T (swap k matches) are reasonable...
> * We apologize for the insufficient explanation on this point. Since defining $d$ directly on images or text can be ambiguous, we instead used the $\ell_2$-norm in the pretrained embedding space.
> * Also, we reported the acceptance ratio of $T$ in **Appendix C.7** along with other MCMC diagnosis results. Acceptance probability of  $\mathbf{T}$ on other datasets such as **Civil** was similar to the reported values ($[0.3, 0.4]$).
>
> ## References
> [1] Wasserstein fair classification. Jiang et. al. UAI. 2020.
>
> [2] A general approach to fairness with optimal transport. Silvia et. al. AAAI. 2020.

---

> ### Author Response · Authors · 2025-11-20
>
> > W4. Complexity claims...
> * Yes, the computational complexity of the MH proposal for the matched Gibbs posterior is $O(n)$ while the Gibbs posterior with MMD penalty is $O(n^2).$
>
> * This advantage really matters when even medium sized data are analyzed. For example, we attempted to run the Gibbs posterior with the MMD penalty on the Adult dataset (whose sample size is $n=45,222$), but they require $O(n^2)$ memory, causing the process to run out of memory and fail in our experimental setup.
>
> * To additionally support our claim, we provide a running time table on **Civil** dataset below (same as Table 9 in Appendix E.5.), with the time measurement on each component. We can observe that the time spent on sampling is relatively small, in the overall algorithm.
>
> **Table 9. A running time table on Civil dataset. Etc. denotes the time measurement on initializations.**
> | Method        | Time (sec.) | Etc. (sec.)                          |
> |------|-------------|------------------------------------- |
> | gapreg        | 48.85       |                                      |
> | reduction     | 47.32       |                                      |
> | gibbs_matched | 24.49       | pretrain: 46.87, ot_initial: 48.56 |
>
> * This is because (1) the scale on each step is small as mentioned, and (2) since we designed the procedure to use a pretrained model as a good initial point for $f$ and an OT-based initialization for $\mathbf{T}$, the proposed MCMC converges quickly and reliably. Additional supporting evidence for the reliability of the proposed MCMC can be found in **Appendix C.7.**
>
> > W5. Concerns about the dataset...
> * Following the reviewer’s concern, we conducted additional experiments on the **ACSIncome** dataset as well. The results are provided in **Appendix E.6.** We can observe that matched Gibbs posterior exhibits comparatively strong performance, similar to other datasets.
>
> > W6. Limitations...
> * For the multinary sensitive attribute case ($S\in\{0,1,\ldots,M\}$ for an integer $M>1$), as mentioned in our paper, the method can be extended by borrowing the approach from paper [3]. More specifically, we may choose one sensitive group (for example, $\mathcal{X}_0$) as an anchor group and define matching functions  $\mathbf{T}_i:\mathcal{X}_i\to\mathcal{X}_0$ that map each of the other sensitive groups to this anchor group.
>
> * In this setting, we can redefine the matched deviation as the sum of matched deviation with respect to $\mathbf{T}_i$, and then construct matched Gibbs posterior using this quantity.
>         For MCMC procedure, the extension is also straightforward: we can simply apply the Gibbs sampler iteratively for each $\mathbf{T}_i$. Please refer to **Appendix E.2** for more detailed discussion on this extension.
>
> * Research on continuous sensitive attributes generally requires additional methodological components. For example, as noted in [4], techniques such as binning or kernel smoothing are commonly introduced to address this setting. By incorporating such additional components, it should be possible to extend the approach used for multinary sensitive attributes to the continuous case as well.
>
> * However, studies on continuous sensitive attributes are often conducted as a separate line of research. For this reason, we believe that extending our framework in this direction goes beyond the intended scope of our work, and it is more appropriate to leave this as a natural direction for future work. Thank you for your suggestions.
>
> * For the discussion regarding EO, please see our response to Weakness 1.
>
> ## References
> [3] Fair clustering via alignment. Kim et. al. ICML. 2025.
>
> [4] Generalized demographic parity for group fairness. Jiang et. al. ICLR. 2022.

---

> > ### Comment · Reviewer_vJDv · 2025-11-21
> > **Reply to the rebuttal**
> >
> > Thanks for the reply.
> >
> > I've read through the improvement you made and I think it's a strong complement to the paper. I highly recommend you to include all these discussion and new experiments into the current draft. I will raise my score, thank you.

---

> > > ### Author Response · Authors · 2025-11-23
> > >
> > > Thank you for your comments and for raising the score. We will further revise the draft as you recommended. Thank you.

---

### Official Review · Reviewer_26og · 2025-11-01

**Soundness:** 2
**Presentation:** 2
**Contribution:** 2
**Rating:** 4
**Confidence:** 4

**Summary:**

This paper addresses the underexplored challenge of integrating model uncertainty into algorithmic fairness. To this end, the authors propose a group-fair posterior distribution and develop a fair variational Bayesian inference framework that embeds fairness constraints into probabilistic learning. To improve computational efficiency, they introduce a novel matched Gibbs posterior, which approximates the fair posterior under fairness constraints using a newly defined metric, matched deviation. This measure is theoretically shown to be closely related to established fairness notions, thereby offering strong fairness guarantees. Empirical evaluations on real-world datasets demonstrate that the proposed approach achieves superior trade-offs between fairness, model uncertainty, and predictive utility compared to baseline methods.

**Strengths:**

1. The topic of group fairness is highly relevant and socially significant, making this study timely and impactful.

2. The authors effectively integrate group fairness and model uncertainty within a Bayesian inference framework and provide solid theoretical justification for their approach.

3. Experiments conducted on three distinct modalities of datasets demonstrate the robustness and practical applicability of the proposed method.

**Weaknesses:**

The paper’s writing and structure could be improved for clarity. After reading, several conceptual issues remain ambiguous:

a. What is the motivation for jointly modeling group fairness and uncertainty? Are there concrete real-world applications that benefit from this combination?

b. What are the main technical challenges in combining fairness and uncertainty? Have there been prior studies exploring this intersection? Why can’t existing fairness and uncertainty methods simply be combined?

c. Why is the Bayesian inference framework particularly appropriate for this setting? What specific advantages does it offer compared to prior non-Bayesian approaches?

The work only considers DP as the fairness definition, which is too limited. Other widely used metrics, such as EO and EOdds, should also be discussed. It remains unclear whether the proposed theory and framework can generalize to these metrics.

In the experimental section, fairness evaluation is restricted to the Wasserstein distance under DP. The empirical validation could be strengthened by including additional fairness metrics to better demonstrate the generalizability and robustness of the proposed method.

**Questions:**

See weaknesses part.

---

> ### Author Response · Authors · 2025-11-20
>
> Thank you for your careful reviews and thoughtful questions. We have done our best to address all of the points and concerns you raised. In the revised paper, we marked the differences and added results in blue. Please note that the figure and section numbers mentioned in the comments correspond to those in the **revised version** of the paper.
>
> # Summary of Changes
> Here is a summary of the changes made in the revised version.
> * Section 1: We revised the introduction to emphasize our motivation and contributions.
> * Appendix E.1: We explain the extension of matched Gibbs posterior with respect to Equalized Odds, along with additional experimental results.
> * Appendix E.2: We explain the extension of matched Gibbs posterior to the case of multinary sensitive cases.
> * Appendix E.3: We report additional experimental results, where the fairness metric is evaluated with different metric, strong demographic parity (SDP).
> * Appendix E.4: We report additional experimental results, when the prior choice varies.
> * Appendix E.5: We further discuss about the computational cost of matched Gibbs posterior.
> * Appendix E.6: We report additional experimental results, on an additional dataset, **ACSIncome**.
>
> # Responses
> > W1. The paper’s writing and structure could be improved for clarity. After reading, several conceptual issues remain ambiguous:
>
> * We would like to emphasize that **uncertainty quantification is essential in decision-making** regardless of whether there is a fairness constraint or not. It is well known that complex prediction model such as deep neural networks are vulnerable to overfitting and so they give over-confident predictions, which should be avoided for critical decision makings such as autonomous driving, medical diagnose and so on. One way to resolve over-confident problems is to quantify uncertainty properly in predictions.
>
> * There are vast amount of literature for proper uncertainty quantification including deep ensemble and Bayesian approaches. We have cited various papers about uncertainty quantification in Introduction of our paper, but all of papers are about uncertainty quantification without fairness constraint, but there extensions for fairness constraints would not be easy due to computational complexity. We believe that **our work is the first of its kinds for uncertainty quantification under fairness constraint.**
>
> * The aim of this paper is to provide proper uncertainty quantification under fairness constraints. The task of learning accurate prediction models under fairness constraints has been an active research topic. However, extending this to include uncertainty quantification is not a simple task: it requires moving from estimation of a single value to estimation at the distribution level, which is substantially more challenging. To the best of our knowledge, there is no prior work for this problem, and our study provides **the first methodological direction** for addressing uncertainty quantification under fairness constraints. For this purpose, we propose a (pseudo) Bayesian approach. It is well known that Bayesian approaches works well for uncertainty quantification but its extension for fairness constraints is computationally challenge since computation of posterior distribution over a constraint parameter space is extremely hard. We avoid this challenge by considering a specially designed Gibbs posterior so called *matched Gibbs posterior*. A novel feature of the matched Gibbs posterior is that we can explore the posterior distribution without any constraint on the parameter space and thus standard MCMC algorithm can be directly used without much hamper. **We added this discussion in Introduction of the revised manuscript.**
>
> * As a real-world application, we can consider the medical decision making problems.
>
>     (1) Suppose one is developing a model for diagnosing skin cancer. It it known that datasets often contain far more images of light-skinned patients, which leads to bias [1]. Therefore, fairness with respect to such data imbalance is required, and at the same time, uncertainty quantification is essential for supporting a physician's decision making.
>
>     (2) Consider the task of developing an Alzheimer’s disease diagnostic model based on brain MRI images. In many datasets, there are plenty of scans from healthy white patients, whereas for other racial groups the proportion of unhealthy cases tends to be higher [2]. This imbalance introduces bias into the data, so fairness becomes a key requirement, and at the same time uncertainty quantification is essential to support the physician’s decision making.
>
> ## References
> [1] Artificial intelligence and skin cancer. Frontiers in medicine. 2024.
>
> [2] Bias in unsupervised anomaly detection in brain MRI. Workshop on Clinical Image-Based Procedures. 2023.
>
> **We added these two references with discussions in Introduction of the revised manuscript.**

---

> ### Author Response · Authors · 2025-11-20
>
> > W2. The work only considers DP as the fairness definition, which is too limited...
> * We focus on DP for the sake of simplicity. However, we can easily extend our proposed matched Gibbs posterior to handle equalized odds. Please refer to **Appendix E.1** for the detailed formulation and corresponding MCMC procedure.
> Using this formulation, we additionally perform an experiment to yield utility-fairness and uncertainty-fairness trade-offs.
> We can continuously observe that matched Gibbs posterior shows better performances than other baselines. Therefore, we believe that the proposed matched Gibbs posterior can also be well applied to equalized odds.
>
> > W3. In the experimental section, fairness evaluation is restricted to the Wasserstein distance under DP...
> * Evaluating DP using the Wasserstein distance has already been explored in prior works [3,4], hence we also considered the Wasserstein distance. However, in response to the reviewer’s concern, we now provide the following trade-off using Strong Demographic Parity (SDP) [3,4], which is one of a representative fairness metric.
> The results are provided in **Appendix E.3.**
> We can consistently observe that the performance of matched Gibbs posterior is better than the competitors.
>
> ## References
> [3] Wasserstein fair classification. Jiang et. al. UAI. 2020.
>
> [4] A general approach to fairness with optimal transport. Silvia et. al. AAAI. 2020.

---

### Meta-Review · Area_Chair_8mfu · 2025-12-18

**Summary:**

The reviewers agree the paper solves the timely problem of integrating group fairness and model uncertainty within a Bayesian inference framework with theoretical justification and strong experiments. There are concerns on the motivation, technical details, fairness definitions, complexity, experiments, and limitations, but the authors thoroughly address them. One reviewer is satisfied, another increased his/her score, and the last one did not respond.

**Reviewer Concerns:**

The authors addressed all the concerns on the motivation, technical details, fairness definitions, complexity, experiments, and limitations where most of the changes went to the Appendix. The AC does not see any concerns that are still outstanding.

**Reviewer Scores:**

* Reviewer 26og: did not respond to rebuttal and has a score of 4
* Reviewer vJDv: based on the discussion, clearly increased the score from 4 to 6
* Reviewer ZbGT: the score is 6

---

### Decision · Program_Chairs · 2026-01-26

Accept (Poster)